# Correlation between the IgG/IgA Antibody Response against PEDV Structural Protein and Virus Neutralization

Xu Song,[a,b,c] Jiali Qian,[a,b] Chuanhong Wang,[a,b] Dandan Wang,[a,b] Junming Zhou,[a,b] Yongxiang Zhao,[a,b] Wei Wang,[a,b] Jizong Li,[a,b] Rongli Guo,[a,b] Yunchuan Li,[a,b] Xuejiao Zhu,[a,b] Shanshan Yang,[a,b] Xuehan Zhang,[a,b] Baochao Fan,[a,b,d,f] Bin Li[a,b,c,e,f]

aInstitute of Veterinary Medicine, Jiangsu Academy of Agricultural Sciences, Key Laboratory of Veterinary Biological Engineering and Technology, Ministry of Agriculture; Jiangsu Key Laboratory for Food Quality and Safety-State Key Laboratory Cultivation Base of Ministry of Science and Technology, Nanjing, China

bJiangsu Coinnovation Center for Prevention and Control of Important Animal Infectious Diseases and Zoonoses, Yangzhou University, Yangzhou, People's Republic of China

cSchool of Veterinary Medicine, Hebei Agricultural University, Baoding, China

dSchool of Life Sciences, Jiangsu University, Zhenjiang, China

eSchool of Food and Biological Engineering, Jiangsu University, Zhenjiang, China

fGuoTai (Taizhou) Center of Technology Innovation for Veterinary Biologicals, Taizhou, China

**ABSTRACT** Porcine epidemic diarrhea (PED) is a highly contagious intestinal infectious disease caused by porcine epidemic diarrhea virus (PEDV). Large-scale outbreaks of PEDV have caused huge economic losses to the pig industry since 2010. Neutralizing antibodies play a pivotal role in protecting piglets from enteric infections. However, there has been no systematic report on the correlations between neutralizing antibody titers (NTs) and absorbance values of IgG or IgA to all PEDV individual structural proteins in clinical serum, fecal, and colostrum samples. In this study, the spike protein S1 domain (S1), membrane protein (M), envelope protein (E), and nucleocapsid protein (N) of the variant PEDV strain AH2012/12 were expressed and purified by using the human embryonic kidney (HEK) 293F expression system. A total of 92 clinical serum samples, 46 fecal samples, and 33 colostrum samples were collected, and the correlations between IgG or IgA absorbance values and NTs were analyzed. $R^2$ values revealed that anti-S1 IgA absorbance values show the highest agreement with NTs in all serum, fecal, and colostrum samples, followed by the N protein. The correlations between anti-E or M IgA and NTs were very low. However, in the colostrum samples, both IgG and IgA to S1 showed high correlations with NTs. In addition, compared with E and M, the highest correlations of IgA absorbance values were with N and S1 in serum and fecal samples. Overall, this study revealed the highest correlation between NTs and IgA to PEDV S1 protein. Therefore, the diagnostic method with anti-S1 IgA can be used as a powerful tool for assessing the immune status of pigs.

**IMPORTANCE** The humoral immune response plays an important role in virus neutralization. Against PEDV, both IgG and the mucosal immune component IgA play roles in virus neutralization. However, which plays a more prominent role and whether there are differences in different tissue samples are not clearly reported. Additionally, the relationship between IgG and IgA against individual structural proteins and viral neutralization remains unclear. In this study, we systematically determined the relationship between IgG and IgA against all PEDV structural proteins and viral neutralization in different clinical samples and found the highest correlation between neutralization activity and IgA to PEDV S1 protein. Our data have important guiding implications in the evaluation of immune protection.

**KEYWORDS** PEDV, IgA, IgG, S1, neutralizing antibody titers

Address correspondence to Baochao Fan, fanbaochao.0405@163.com, or Bin Li, libinana@126.com.

The authors declare no conflict of interest.

Porcine epidemic diarrhea (PED) is an acute diarrhea, dehydration, and high-mortality digestive tract disease caused by the porcine epidemic diarrhea virus (PEDV) (1). It is mainly manifested by vomiting, watery diarrhea, and dehydration, and the mortality rate in piglets with diarrhea is up to 90 to 100% (2, 3). PEDV infects pigs mainly via the fecal-oral route; therefore, protective mucosal immunity is essential for preventing infection. PEDV was first discovered in the United Kingdom in 1971 and was later found in other European countries (4). In October 2010, a widespread outbreak of PED caused by highly virulent PEDV variants distinct from the classic strain CV777 occurred in China (5). In 2013, new PEDV variants containing new insertions and deletions in the S gene versus the prototype strain were reported in the United States (6). Subsequently, these variants, named S-indel-variants, were also detected and isolated in China (7). Phylogenetic studies suggested that PEDV can be genetically separated into G1 (classic), G2 (field epidemic), and S-indel-like genotypes. The G1 and G2 genotypes can be further divided into the subgenotypes G1a and G1b and G2a and G2b, respectively (8). Recent studies revealed that the G2a subgroup has been increasing rapidly and has become dominant in recent years (9).

PEDV is a member of the genus alphacoronavirus of the family *Coronaviridae*. It is an enveloped, single-stranded, positive-sense RNA virus with a 28-kb genome that encodes several nonstructural replicase proteins and four major structural proteins, including the spike (S), envelope (E), membrane (M), and nucleocapsid (N) proteins and an accessory open reading frame 3 (ORF3) between S and E (10). Based on the homology of other coronaviruses, the S protein of PEDV is divided into two subunits, S1 and S2, which are the main surface glycoproteins involved in viral attachment, entry, and induction of a protective immune response (11). The N protein is also multifunctional. In the early stage of PEDV infection, high levels of anti-N protein antibodies can appear in pigs, and N protein is highly conserved and can be used as a target protein for early diagnosis (12). M protein is a highly conserved transmembrane protein that plays an important role in viral assembly, germination, and host immune regulation (13, 14). Similar to other structural proteins, E protein is also a versatile protein involved in the formation of viral capsule membranes and plays an important role in viral replication and germination (15).

PEDV stimulates and induces local responses in the body's intestinal mucosa, in which antibody-secreting cells play important roles in mucosa-associated lymphoid tissue and serum antibodies (16). Several studies have found that the combined action of PEDV-specific IgG and IgA produced *in vivo* after sow infection caused high levels of neutralization titers (17, 18). There was a positive correlation between sera neutrality activity in piglets after PEDV infection and lower clinical scores. At present, a few studies have reported the correlation between IgA induced by PEDV S protein and neutralizing activity. Song et al. found that the neutralizing activity significantly correlated with specific IgA, primarily to S1 protein of PEDV rather than to specific IgG in colostrum (19), and San et al. indicated that sera with higher neutralizing activity had a higher IgA antibody level based on S1 established indirect enzyme-linked immunosorbent assay (ELISA) (20). However, for other structural proteins E, M, and N of PEDV, relationships between the antibody levels induced and neutralizing activities and the roles of IgG and IgA induced by all structural proteins in viral neutralization activities are still unclear.

In this study, the S1, E, M, and N proteins of the variant PEDV strain AH2012/12 were expressed and purified by using the human embryonic kidney (HEK) 293F expression system. The IgA and IgG antibodies to these proteins in clinical serum, fecal, and colostrum samples were detected by indirect ELISA methods, and correlations between the IgA and IgG optical density (OD) values for S1, E, M, and N and the neutralizing antibody titers (NTs) were determined. These data provide an important reference index for evaluating the immune status of PEDV-infected or vaccinated pigs.

## RESULTS

**Expression and purification of viral structural proteins.** The expression of recombinant plasmids was determined *in vitro* by indirect immunofluorescence (IF) assays. As shown

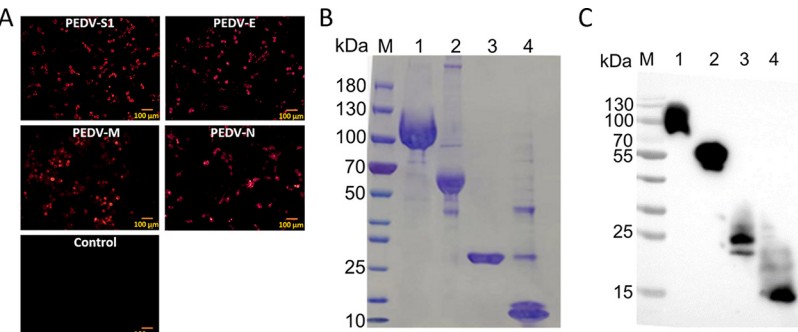

**FIG 1** Expression and purification of the PEDV S1, E, M, and N proteins. (A) IF assays of 293T cells transfected with eukaryotic plasmids pPEDV-S1, pPEDV-E, pPEDV-M, or pPEDV-N. (B) SDS-PAGE analyses of purified PEDV S1, E, M, and N according to the Materials and Methods. (C) Western blotting of purified PEDV S1, E, M, and N proteins using anti-His tag antibody; scale bars, 100 $\mu$m; M, protein maker; 1, pPEDV-S1; 2, pPEDV-N; 3, pPEDV-M; 4, pPEDV-E.

in Fig. 1A, the specific fluorescence of PEDV S1, E, M, and N with high expression was observed at 24 h posttransfection. To establish the indirect ELISA methods, these proteins were expressed in 293F cells and purified with HisTrap HP columns. SDS-PAGE and immunoblotting were used to verify the purities of these purified proteins, and the theoretical molecular weights of PEDV S1, E, M, and N were 100, 50, 25, and 11 kDa, respectively. As shown in Fig. 1B, the SDS-PAGE assay showed bands with the expected molecular masses and high purity after staining with Coomassie brilliant blue. The purified proteins were further validated by immunoblotting (Fig. 1C), and the bands were identical to the bands visualized by SDS-PAGE. The protein concentrations measured using a bicinchoninic acid (BCA) kit (Beyotime, China) were 0.8, 0.2, 0.2, and 0.2 mg/mL, respectively. These results show that the four viral antigens were purified successfully and are suitable to establish the indirect ELISA methods.

**Correlation between IgG/IgA OD values and NT assays in sera.** The NTs of 92 clinical serum samples were determined, with the titers ranging from $2^1$ to $2^8$. The correlations of NT with IgG OD values for S1, E, M, and N in sera were analyzed first. As shown in Fig. 2A, among the correlation coefficients of IgG OD values to NTs for all proteins, the highest $R^2$ was 0.292 for S1 protein. For E, M, and N, the correlation coefficients were 0.058, 0.077, and 0.1356, respectively. Correlations between IgA and NT in sera were also determined and are shown in Fig. 2B. The correlation coefficients of IgA OD values to NT for S1, E, M, and N were 0.7027, 0.123, 0.2002, and 0.426, respectively. The highest agreement was between S1 IgA and NT, and the correlation coefficient of N IgA OD values to NT was also higher than all indexes of IgG OD values of E, M, and N proteins to NT.

**Correlations between IgG/IgA OD values and NT in fecal samples.** The NTs of 46 clinical fecal samples were determined, with the titers ranging from $2^1$ to $2^8$. All samples were further tested for IgG and IgA antibody levels to the S1, E, M, and N proteins. Correlation analyses of NTs and antibody OD values were performed and are shown in Fig. 3. For the fecal samples, the correlation coefficients of IgG OD values to NT for S1, E, M, and N were 0.3441, 0.132, 0.1984, and 0.156, respectively (Fig. 3A), and the correlation coefficients of IgA OD values to NT for S1, E, M, and N were 0.657, 0.151, 0.1982, and 0.313, respectively (Fig. 3B). S1 had the highest correlation with NT versus the other proteins, either IgG or IgA. Similar to the results obtained with sera, the IgA levels of S1 also revealed the highest correlation with NT, followed by the N protein.

**Correlations between IgG/IgA OD values and NT in colostrum.** Because the highest correlation was between antibody levels to S1 and NT, we decided to test reactivity to S1 in colostrum samples. The NTs of 32 colostrum samples were determined, with titers ranging from 1:2 to 1:256. The correlation analysis showed that the correlation coefficients of IgG and IgA OD values to NT for S1 were 0.682 and 0.8477, respectively (Fig. 4). These data indicate that both IgG and IgA to S1 show high correlations with NT in colostrum.

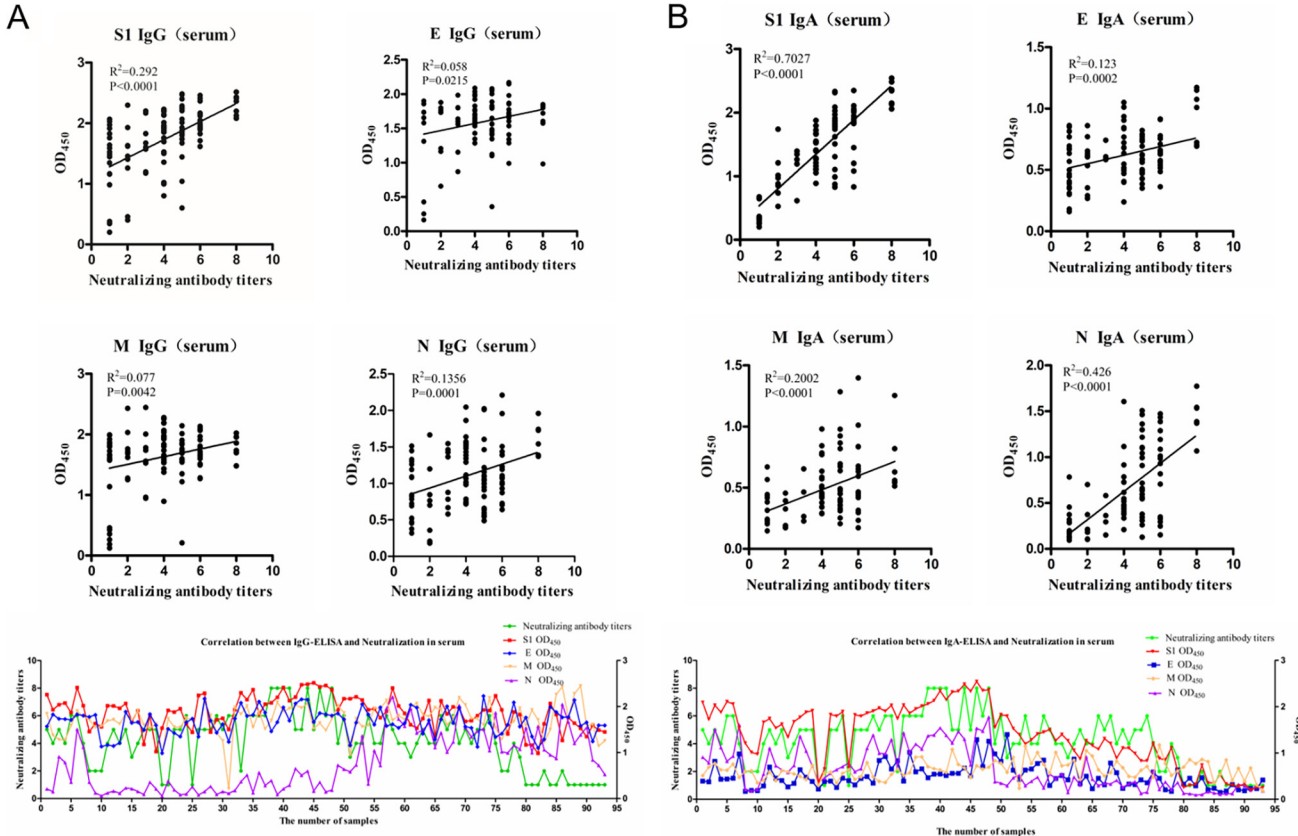

**FIG 2** (A and B) Correlations of NTs with IgG (A) or IgA (B) OD values for S1, E, M, and N in sera. Correlations are shown in scatterplots and line charts. For scatterplots, the *x* axis indicates the NTs, and the *y* axis indicates the antibody OD values. For line charts, the *x* axis indicates the number of samples, the left axis indicates NTs, and the right axis indicates the antibody OD values. $R^2$ values represent the correlation coefficients.

**Correlations of IgA OD values of E, M, and N with IgA OD values of S1 in serum and fecal samples.** The above results confirm that, in addition to the S1 protein, the IgA OD values of N protein are also highly correlated with NT. However, because the N protein is located within the viral envelope, the antibody against N protein has little chance to bind to the N protein of intact virions. So, correlations of IgA OD values of N to S1 in serum and fecal samples were analyzed while also comparing with membrane proteins E and M.

As shown in Fig. 5A and B, the correlation coefficients of the IgA OD values of E, M, and N to the IgA OD values of S1 were 0.04, 0.352, and 0.661 in serum samples and 0.107, 0.103, and 0.641 in fecal samples, respectively. The data show that the highest correlation was between the IgA OD values of N and S1 and revealed that IgA production to N and S1 has a very similar growth and decline timeline.

## DISCUSSION

PEDV S glycoprotein (S1 and S2 domains) containing core neutralizing epitopes and B cell-neutralizing epitopes can stimulate the production of virus-neutralizing antibodies (21). It has been generally known that neutralizing activity significantly correlates with specific IgA primarily to S protein of PEDV rather than to specific IgG in colostrum (19, 22). In this study, we also found that anti-S1 IgA values have the highest correlation with NT in sera, fecal matter, and colostrum. It is worth noting that IgA to N protein also has a relatively high correlation with NT, with correlation coefficients of 0.426 and 0.313 in serum and fecal samples, respectively. Subsequent analysis revealed a very high correlation between the IgA OD values of N and S1 proteins. Such results suggest a high temporal consistency in the production of IgA antibodies against N and S proteins in pigs. This finding also suggests that the measurement of N protein IgA will determine the immune status of the herd to some extent.

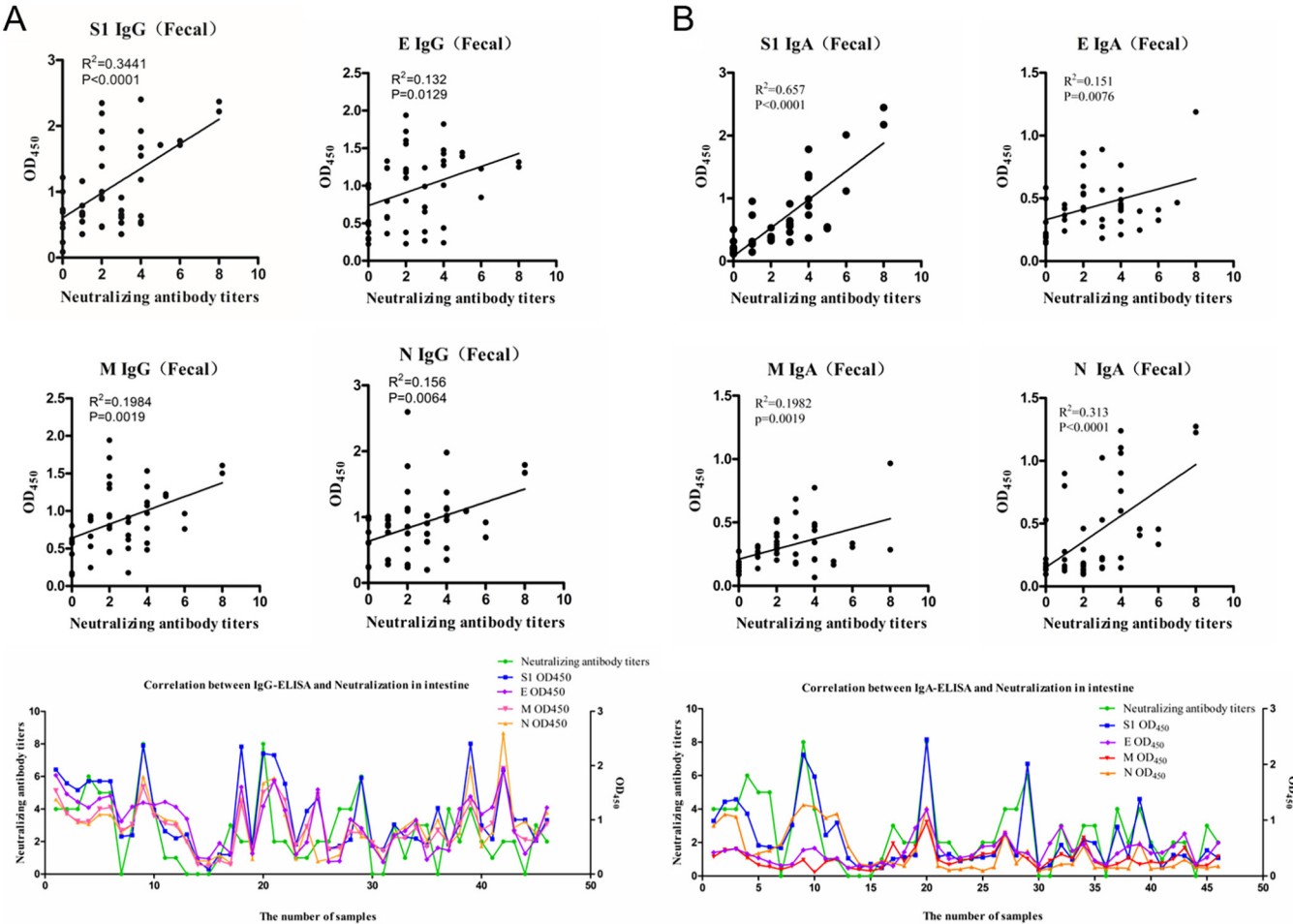

**FIG 3** (A and B) Correlations of NTs with IgG (A) or IgA (B) OD values for S1, E, M, and N in fecal samples. Correlations are shown in scatterplots and line charts. For scatterplots, the $x$ axis indicates the NTs, and the $y$ axis indicates the antibody OD values. For line charts, the $x$ axis indicates the number of samples, the left axis indicates NTs, and the right axis indicates the antibody OD values. $R^2$ values represent the correlation coefficients.

In performing a neutralizing antibody assay, specific antibodies are required to bind the target proteins, thus exerting a blocking effect on the binding of the virus to the host. In previous studies, only nonneutralizing epitopes have been identified in the PEDV N protein (23, 24). This may be related to the presence of N protein in the viral envelope. Of course, it is also important to note that several studies have found that coronavirus N proteins have T cell immune epitopes and play important roles in protective immune responses (25). For PEDV, identification of the T cell epitopes of the N protein should also not be neglected.

Through the gut-mammary-secretory IgA (sIgA) axis, enteric virus-infected sows produce IgA immunocytes, which migrate to the mammary glands and create high titers of sIgA antibodies in the colostrum, supporting passive lactogenic immunity (26, 27). Therefore, it is widely acknowledged that lactogenic immunity, or anti-PEDV secretory IgA in milk, is important for inhibition of PEDV replication in intestines and prevention of clinical disease in piglets. In this study, we detected two antibody isotypes, IgG and IgA, to PEDV S1, E, M, and N proteins and analyzed correlations with NT. The results revealed that levels of IgG correlated poorly with neutralizing activity, even for IgG antibodies to S1 protein in serum and fecal samples.

In this study, we found weak correlations between S1 IgG antibody levels in serum and fecal samples and NT. However, S1 IgG and IgA levels in colostrum are highly correlated with NT. One major reason for this difference is most likely related to the fact that colostrum contains large amounts of immunoglobulins. The correlation between S1 IgG and NT increases due to the presence of large amounts of IgG and IgA. This was also observed

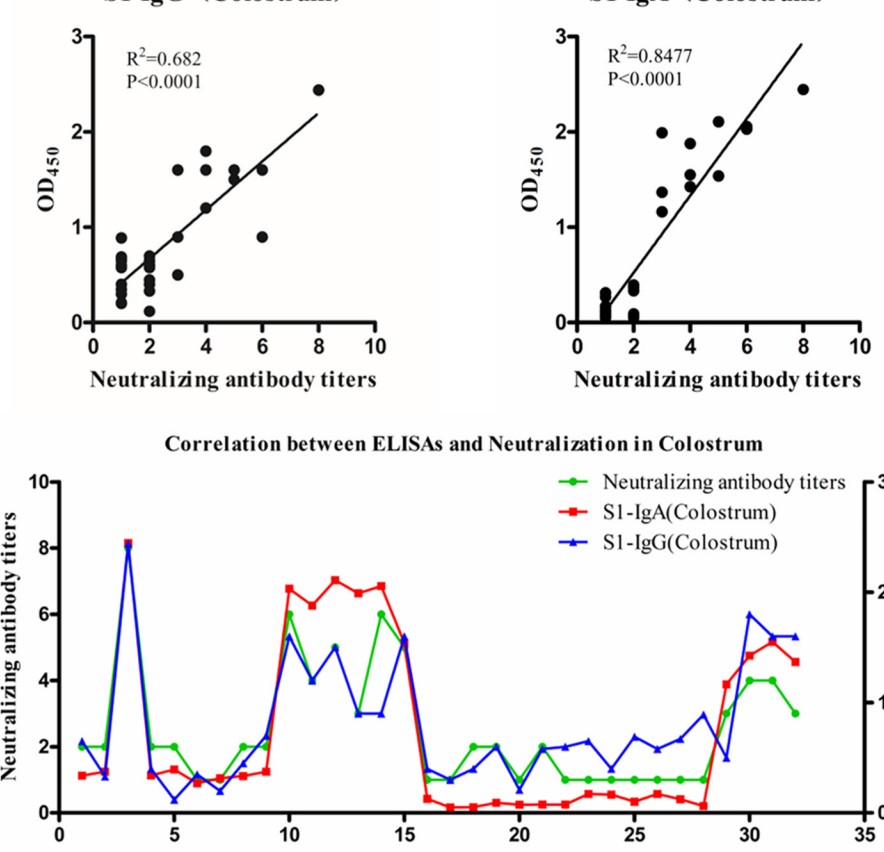

**FIG 4** Correlations of NTs with IgG or IgA OD values for S1 in colostrum samples. Correlations are shown in scatterplots and line charts. For scatterplots, the *x* axis indicates the NTs, and the *y* axis indicates the antibody OD values. For line charts, the *x* axis indicates the number of samples, the left axis indicates NTs, and the right axis indicates the antibody OD values. $R^2$ values represent the correlation coefficients.

in a previous study. Song et al. reported that virus NTs and levels of total IgG and IgA against S1 in colostrum significantly increased in milk at days 1 to 3 and declined progressively. In addition, from days 3 to 19, the IgG levels decreased sharply while IgA levels declined slightly, more like the trend of virus-neutralizing activities at the same time (19). The colostrum samples used in this study were also collected within 1 to 3 days of delivery in sows.

A neutralization test is a fundamental method for PEDV antibody detection in pigs with prior infection. However, there are many restrictions with the requirements of technical expertise, aseptic conditions, and time-consuming tests (26). In comparison, ELISAs, which permit more convenient and rapid analysis, have been developed for PEDV antibody detection. With the highest correlation between NT and IgA to PEDV S1 protein, the S protein is therefore well regarded as a suitable candidate for neutralizing antibody detection, as described before (28). This diagnostic method can be used not only for diagnosis of clinical viral infection but also for evaluation of the effect after vaccine immunization. Of course, a high level of genetic diversity of S proteins between different PEDV strains may limit its usability as an antibody detection platform in field applications. However, considering the current clinical epidemiology of PEDV, the G2 variant strain remains the most prevalent strain and is likely to remain so for a long time; so, the anti-S IgA ELISA method is currently still a very effective diagnostic tool for disease. It needs to be mentioned that the S protein, including S1 and S2, possesses multiple neutralizing epitopes (21). Before the study, we attempted to use the full-length S protein for the assays, but the purified protein

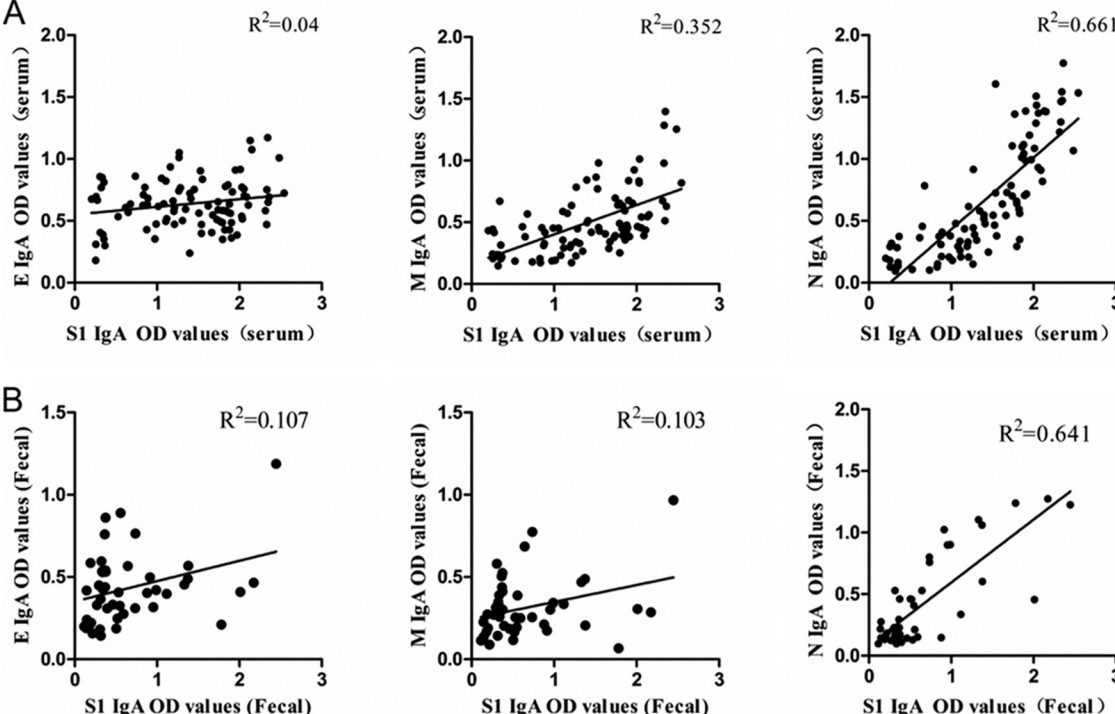

**FIG 5** (A and B) Correlations of IgA OD values of E, M, and N with IgA OD values of S1 in serum and fecal samples. Correlations of the IgA OD values of E, M, and N to the IgA OD values of S1 in serum (A) and fecal (B) samples are shown. Correlations are shown in scatterplots. $R^2$ values represent the correlation coefficients.

was poorly effective and difficult to meet the requirements of the tests. As S1 contains the core neutralizing and several antigenic epitopes (21), we selected S1 with high purity for subsequent studies. In the future, the assays will be performed using high-purity full-length S, enriching the study data.

In our screening studies of monoclonal antibodies against PEDV S proteins, we found that antibodies with neutralizing effects tend to target conformational epitopes, and the vast majority of antibodies to linear epitopes do not have neutralizing activity (data not shown). Eukaryotic expression systems have the unique advantage of being able to produce natural conformational proteins with appropriate posttranslational modifications (8), Therefore, in this study, we used the 293F expression system to express S1, E, M, and N proteins. The near-natural conformation of the S1 protein may be the key to the high correlation with neutralization titers.

In conclusion, the anti-S1 IgA OD values show the highest agreement with NT in all serum, fecal, and colostrum samples, and the anti-S1 IgA ELISA can be used as a tool to predict the neutralization capacity of clinical sera, feces, and colostrum and to evaluate the infection and immune status of pigs.

## MATERIALS AND METHODS

**Cells and viruses.** Vero-81 (ATCC, CCL-81) cells, which were used for propagation of PEDV, were cultured in Dulbecco's modified Eagle's medium (DMEM) supplemented with 5% (vol/vol) fetal bovine serum (FBS) at 37℃ and 5% $CO_2$ (29–32). PEDV strain AH2012/12 (GenBank accession number KU646831) was isolated and preserved in our laboratory, as described previously (33). The human embryonic kidney (HEK) 293T cells preserved in our lab were maintained in DMEM containing 10% FBS at 37℃ and 5% $CO_2$. The HEK 293F cells were purchased from Thermo Fisher Scientific and cultured in serum-free SMM293-TII medium (Sino Biological, Inc., China) in suspension.

**Clinical samples.** Ninety-two clinical serum samples, 46 fecal samples from piglets or fattening pigs, and 32 colostrum samples from sows with or without diarrhea were collected from pig farms in Jiangsu province. Serum samples were all collected from fattening pigs. For the fecal samples, 32 samples were collected from piglets within 1 month, 11 from fattening pigs, and 3 from sows. The colostrum samples were collected within 1 to 3 days of delivery in sows. The details of the samples are shown in Table 1.

**TABLE 1** Antibody OD values and NT results of each sample

| Samples | No. | Collection | Different proteins, IgG values | | | | Different proteins, IgA values | | | | NTs |
|---|---|---|---|---|---|---|---|---|---|---|---|
| | | | S1 | E | M | N | S1 | E | M | N | |
| Sera | 1 | All samples from | 2.257 | 1.568 | 1.844 | 0.209 | 2.090 | 0.392 | 0.511 | 0.909 | 5 |
| | 2 | fattening pigs | 1.929 | 1.825 | 1.375 | 0.150 | 1.725 | 0.378 | 0.693 | 0.785 | 4 |
| | 3 | | 2.039 | 1.732 | 1.279 | 0.909 | 2.032 | 0.820 | 0.773 | 1.507 | 5 |
| | 4 | | 2.075 | 1.727 | 1.271 | 0.785 | 1.879 | 0.432 | 0.636 | 1.116 | 4 |
| | 5 | | 1.869 | 1.707 | 1.586 | 0.357 | 2.110 | 0.437 | 0.628 | 0.820 | 6 |
| | 6 | | 2.412 | 1.829 | 1.905 | 1.507 | 2.056 | 0.454 | 0.758 | 0.931 | 6 |
| | 7 | | 2.019 | 1.803 | 1.781 | 1.116 | 1.538 | 0.980 | 0.471 | 1.605 | 4 |
| | 8 | | 1.456 | 1.775 | 1.714 | 0.357 | 1.211 | 0.172 | 0.607 | 0.208 | 2 |
| | 9 | | 1.632 | 1.817 | 1.544 | 0.121 | 1.013 | 0.192 | 0.618 | 0.180 | 2 |
| | 10 | | 1.560 | 1.129 | 1.573 | 0.058 | 0.974 | 0.187 | 0.355 | 0.208 | 2 |
| | 11 | | 2.017 | 1.163 | 1.687 | 0.122 | 1.663 | 0.288 | 0.406 | 0.544 | 4 |
| | 12 | | 2.012 | 1.142 | 1.717 | 0.162 | 1.741 | 0.696 | 0.590 | 1.105 | 5 |
| | 13 | | 1.961 | 1.208 | 1.62 | 0.116 | 1.539 | 0.457 | 0.399 | 0.726 | 4 |
| | 14 | | 2.065 | 1.887 | 1.856 | 0.229 | 1.620 | 0.405 | 0.620 | 0.636 | 5 |
| | 15 | | 1.673 | 1.645 | 1.696 | 0.212 | 1.346 | 0.266 | 0.594 | 0.581 | 3 |
| | 16 | | 1.916 | 1.660 | 1.912 | 0.154 | 1.676 | 0.343 | 0.548 | 0.729 | 4 |
| | 17 | | 1.176 | 1.486 | 1.599 | 0.088 | 1.771 | 0.639 | 0.494 | 1.363 | 5 |
| | 18 | | 1.818 | 1.834 | 1.528 | 0.356 | 1.882 | 0.477 | 0.427 | 1.042 | 5 |
| | 19 | | 1.022 | 1.592 | 1.624 | 0.208 | 1.913 | 0.385 | 0.734 | 0.718 | 5 |
| | 20 | | 1.531 | 0.985 | 0.965 | 0.052 | 0.367 | 0.219 | 0.300 | 0.375 | 1 |
| | 21 | | 1.917 | 1.512 | 1.734 | 0.385 | 0.650 | 0.383 | 0.569 | 0.456 | 1 |
| | 22 | | 1.879 | 1.423 | 1.552 | 0.190 | 1.831 | 0.392 | 0.583 | 0.561 | 5 |
| | 23 | | 1.950 | 1.345 | 1.547 | 0.082 | 1.797 | 0.253 | 0.351 | 0.640 | 5 |
| | 24 | | 1.617 | 1.656 | 1.752 | 0.099 | 1.891 | 0.456 | 0.561 | 0.706 | 6 |
| | 25 | | 1.441 | 1.734 | 1.713 | 0.255 | 0.677 | 0.401 | 0.636 | 0.784 | 1 |
| | 26 | | 2.287 | 2.170 | 2.099 | 0.152 | 1.823 | 0.310 | 0.427 | 0.594 | 5 |
| | 27 | | 1.444 | 1.693 | 1.809 | 0.263 | 1.795 | 0.450 | 0.486 | 0.659 | 5 |
| | 28 | | 1.711 | 1.431 | 1.435 | 0.087 | 1.861 | 0.491 | 0.790 | 1.099 | 5 |
| | 29 | | 1.743 | 1.616 | 1.255 | 0.152 | 1.905 | 0.349 | 0.486 | 1.388 | 6 |
| | 30 | | 1.500 | 1.231 | 0.211 | 0.113 | 1.971 | 0.837 | 0.386 | 0.994 | 5 |
| | 31 | | 1.929 | 1.737 | 1.986 | 0.174 | 1.903 | 0.924 | 0.363 | 0.985 | 6 |
| | 32 | | 2.298 | 2.08 | 1.835 | 0.366 | 2.030 | 0.654 | 0.538 | 1.290 | 6 |
| | 33 | | 2.149 | 1.792 | 2.013 | 0.512 | 1.742 | 0.834 | 0.654 | 0.701 | 2 |
| | 34 | | 2.366 | 1.819 | 1.547 | 0.126 | 2.036 | 0.393 | 0.516 | 1.435 | 6 |
| | 35 | | 1.826 | 2.106 | 1.883 | 0.566 | 1.857 | 1.012 | 0.572 | 1.013 | 6 |
| | 36 | | 2.050 | 1.867 | 1.511 | 0.282 | 1.952 | 0.594 | 0.910 | 1.192 | 6 |
| | 37 | | 2.081 | 1.46 | 1.692 | 0.391 | 2.011 | 0.655 | 0.915 | 1.086 | 6 |
| | 38 | | 2.199 | 1.659 | 2.093 | 0.086 | 2.058 | 0.465 | 0.703 | 1.370 | 8 |
| | 39 | | 2.414 | 1.98 | 1.899 | 0.211 | 2.131 | 0.542 | 0.661 | 1.387 | 8 |
| | 40 | | 2.126 | 1.693 | 1.957 | 0.289 | 2.344 | 0.546 | 0.635 | 1.543 | 8 |
| | 41 | | 2.206 | 2.065 | 2.033 | 0.650 | 2.150 | 0.513 | 0.583 | 1.382 | 8 |
| | 42 | | 2.471 | 2.157 | 1.967 | 0.505 | 2.315 | 0.560 | 1.076 | 1.217 | 5 |
| | 43 | | 2.484 | 2.148 | 2.065 | 0.595 | 2.334 | 0.561 | 0.752 | 1.464 | 5 |
| | 44 | | 2.515 | 1.664 | 1.513 | 0.161 | 2.484 | 0.674 | 0.679 | 1.299 | 8 |
| | 45 | | 2.430 | 1.855 | 1.641 | 0.355 | 2.332 | 0.502 | 0.471 | 0.742 | 5 |
| | 46 | | 2.458 | 1.83 | 1.480 | 0.146 | 2.543 | 1.286 | 0.724 | 1.534 | 8 |
| | 47 | | 2.363 | 1.802 | 1.735 | 0.184 | 2.348 | 0.704 | 0.651 | 1.472 | 6 |
| | 48 | | 1.930 | 1.849 | 2.024 | 0.726 | 2.362 | 1.255 | 0.693 | 1.775 | 8 |
| | 49 | | 2.169 | 1.721 | 1.859 | 0.636 | 1.552 | 0.979 | 0.836 | 0.379 | 4 |
| | 50 | | 2.169 | 1.154 | 0.946 | 0.581 | 1.830 | 0.819 | 0.778 | 0.349 | 6 |
| | 51 | | 2.211 | 1.648 | 1.738 | 0.729 | 1.796 | 1.398 | 0.584 | 0.294 | 3 |
| | 52 | | 2.141 | 2.052 | 1.559 | 1.363 | 1.527 | 0.629 | 0.905 | 0.465 | 4 |
| | 53 | | 1.930 | 1.829 | 1.591 | 0.318 | 1.396 | 0.429 | 0.239 | 0.517 | 4 |
| | 54 | | 1.950 | 1.606 | 1.961 | 1.042 | 1.194 | 0.633 | 1.149 | 0.295 | 6 |
| | 55 | | 1.757 | 1.981 | 1.957 | 0.718 | 1.283 | 0.654 | 0.768 | 0.389 | 4 |
| | 56 | | 1.897 | 1.576 | 2.134 | 1.960 | 1.411 | 0.771 | 1.010 | 0.478 | 4 |
| | 57 | | 2.401 | 1.615 | 1.698 | 2.212 | 1.450 | 0.843 | 0.755 | 0.247 | 6 |
| | 58 | | 2.012 | 1.536 | 1.804 | 1.729 | 1.508 | 0.297 | 0.562 | 0.546 | 5 |
| | 59 | | 1.852 | 1.639 | 1.715 | 1.571 | 1.267 | 0.447 | 1.012 | 0.444 | 4 |
| | 60 | | 1.603 | 2.037 | 2.007 | 1.957 | 1.400 | 0.513 | 0.722 | 0.548 | 4 |
| | 61 | | 1.527 | 1.977 | 2.065 | 1.426 | 1.264 | 0.419 | 0.742 | 0.150 | 3 |

## TABLE 1 (Continued)

| Samples | No. | Collection | Different proteins, IgG values | | | | Different proteins, IgA values | | | | NTs |
|---|---|---|---|---|---|---|---|---|---|---|---|
| | | | S1 | E | M | N | S1 | E | M | N | |
| | 62 | | 1.737 | 1.410 | 1.835 | 1.481 | 1.055 | 0.866 | 0.844 | 0.480 | 4 |
| | 63 | | 2.131 | 1.702 | 1.780 | 1.639 | 0.873 | 0.340 | 0.429 | 0.309 | 5 |
| | 64 | | 1.356 | 1.284 | 1.977 | 1.108 | 1.268 | 0.498 | 1.051 | 0.915 | 4 |
| | 65 | | 2.123 | 1.718 | 1.935 | 1.431 | 1.162 | 0.227 | 0.935 | 0.435 | 4 |
| | 66 | | 1.840 | 2.004 | 1.600 | 1.011 | 1.081 | 0.453 | 0.638 | 0.295 | 6 |
| | 67 | | 1.994 | 1.640 | 1.761 | 1.230 | 1.309 | 0.312 | 0.502 | 0.345 | 5 |
| | 68 | | 1.962 | 1.976 | 2.192 | 2.026 | 1.203 | 0.784 | 0.577 | 0.329 | 6 |
| | 69 | | 1.675 | 1.562 | 2.035 | 1.544 | 0.890 | 0.572 | 0.822 | 0.409 | 4 |
| | 70 | | 1.695 | 1.536 | 1.739 | 1.340 | 1.106 | 0.233 | 0.683 | 0.261 | 5 |
| | 71 | | 1.801 | 1.123 | 1.736 | 1.464 | 1.106 | 0.329 | 0.495 | 0.336 | 4 |
| | 72 | | 1.926 | 2.229 | 1.597 | 1.511 | 0.989 | 0.316 | 0.475 | 0.379 | 5 |
| | 73 | | 1.913 | 1.642 | 2.188 | 1.040 | 0.830 | 0.418 | 0.626 | 0.127 | 5 |
| | 74 | | 2.232 | 1.339 | 1.535 | 1.864 | 0.833 | 0.356 | 0.711 | 0.153 | 6 |
| | 75 | | 1.928 | 1.410 | 1.285 | 1.017 | 1.211 | 0.587 | 0.472 | 0.615 | 4 |
| | 76 | | 1.407 | 1.606 | 1.572 | 0.992 | 0.882 | 0.452 | 1.172 | 0.212 | 2 |
| | 77 | | 1.883 | 2.089 | 1.673 | 1.157 | 0.852 | 0.206 | 0.771 | 0.374 | 2 |
| | 78 | | 1.829 | 1.563 | 1.961 | 1.070 | 1.115 | 0.172 | 0.517 | 0.209 | 4 |
| | 79 | | 1.167 | 1.760 | 1.584 | 1.541 | 0.617 | 0.636 | 0.608 | 0.364 | 3 |
| | 80 | | 1.160 | 1.347 | 1.925 | 0.998 | 0.258 | 0.327 | 0.696 | 0.128 | 1 |
| | 81 | | 0.983 | 1.100 | 1.709 | 1.569 | 0.305 | 0.455 | 0.857 | 0.120 | 1 |
| | 82 | | 1.622 | 1.286 | 1.219 | 1.059 | 0.322 | 0.292 | 0.770 | 0.094 | 1 |
| | 83 | | 2.065 | 1.992 | 1.497 | 0.950 | 0.736 | 0.464 | 0.861 | 0.103 | 2 |
| | 84 | | 1.827 | 1.879 | 2.259 | 0.872 | 0.355 | 0.206 | 0.813 | 0.164 | 1 |
| | 85 | | 1.262 | 1.764 | 2.431 | 2.046 | 0.353 | 0.237 | 0.352 | 0.136 | 1 |
| | 86 | | 1.759 | 1.769 | 1.762 | 1.666 | 0.526 | 0.147 | 0.534 | 0.108 | 2 |
| | 87 | | 1.638 | 2.139 | 2.278 | 1.199 | 0.330 | 0.179 | 0.850 | 0.119 | 1 |
| | 88 | | 1.527 | 1.581 | 2.447 | 1.384 | 0.309 | 0.316 | 0.405 | 0.124 | 1 |
| | 89 | | 1.346 | 1.655 | 1.585 | 1.547 | 0.274 | 0.204 | 0.668 | 0.319 | 1 |
| | 90 | | 1.589 | 1.242 | 1.615 | 0.84 | 0.340 | 0.185 | 0.389 | 0.284 | 1 |
| | 91 | | 1.487 | 1.591 | 1.14 | 0.721 | 0.203 | 0.236 | 0.676 | 0.198 | 1 |
| | 92 | | 1.447 | 1.589 | 1.254 | 0.523 | 0.256 | 0.230 | 0.310 | 0.292 | 1 |
| Feces | 1 | Fattening pig | 1.925 | 1.821 | 1.534 | 1.377 | 0.988 | 0.421 | 0.343 | 0.903 | 4 |
| | 2 | Fattening pig | 1.673 | 1.476 | 1.116 | 1.121 | 1.331 | 0.454 | 0.469 | 1.104 | 4 |
| | 3 | Fattening pig | 1.548 | 1.333 | 0.971 | 0.956 | 1.374 | 0.489 | 0.488 | 1.061 | 4 |
| | 4 | Sows | 1.712 | 1.226 | 0.966 | 0.921 | 1.116 | 0.398 | 0.335 | 0.335 | 6 |
| | 5 | Piglet | 1.714 | 1.392 | 1.197 | 1.098 | 0.547 | 0.326 | 0.195 | 0.407 | 5 |
| | 6 | Fattening pig | 1.711 | 1.443 | 1.229 | 1.092 | 0.519 | 0.248 | 0.165 | 0.457 | 5 |
| | 7 | Piglet | 0.694 | 0.972 | 0.801 | 0.771 | 0.503 | 0.187 | 0.117 | 0.573 | 0 |
| | 8 | Fattening pig | 0.717 | 1.241 | 0.914 | 0.907 | 0.913 | 0.211 | 0.175 | 1.024 | 3 |
| | 9 | Sows | 2.371 | 1.317 | 1.608 | 1.795 | 2.173 | 0.465 | 0.286 | 1.274 | 8 |
| | 10 | Fattening pig | 1.187 | 1.274 | 1.072 | 1.138 | 1.782 | 0.498 | 0.067 | 1.240 | 4 |
| | 11 | Piglet | 0.791 | 1.330 | 0.930 | 1.008 | 0.733 | 0.310 | 0.256 | 1.041 | 1 |
| | 12 | Piglet | 0.660 | 1.238 | 0.895 | 0.962 | 0.954 | 0.317 | 0.302 | 1.115 | 1 |
| | 13 | Piglet | 0.732 | 1.018 | 0.594 | 0.609 | 0.317 | 0.143 | 0.143 | 0.530 | 0 |
| | 14 | Piglet | 0.234 | 0.301 | 0.176 | 0.245 | 0.138 | 0.189 | 0.118 | 0.218 | 0 |
| | 15 | Piglet | 0.089 | 0.285 | 0.147 | 0.241 | 0.216 | 0.157 | 0.089 | 0.183 | 0 |
| | 16 | Piglet | 0.358 | 0.571 | 0.247 | 0.350 | 0.146 | 0.241 | 0.137 | 0.277 | 1 |
| | 17 | Piglet | 0.358 | 0.389 | 0.178 | 0.202 | 0.306 | 0.182 | 0.581 | 0.229 | 3 |
| | 18 | Piglet | 2.350 | 1.603 | 1.303 | 1.387 | 0.338 | 0.427 | 0.258 | 0.184 | 2 |
| | 19 | Piglet | 0.476 | 0.380 | 0.452 | 0.279 | 0.374 | 0.861 | 0.525 | 0.460 | 2 |
| | 20 | Fattening pig | 2.221 | 1.250 | 1.503 | 1.673 | 2.446 | 1.188 | 0.967 | 1.226 | 8 |
| | 21 | Piglet | 2.193 | 1.723 | 1.709 | 1.773 | 0.339 | 0.529 | 0.290 | 0.182 | 2 |
| | 22 | Piglet | 1.665 | 1.176 | 1.361 | 1.095 | 0.393 | 0.309 | 0.203 | 0.109 | 2 |
| | 23 | Piglet | 0.548 | 0.364 | 0.530 | 0.283 | 0.268 | 0.329 | 0.271 | 0.124 | 1 |
| | 24 | Piglet | 1.163 | 0.587 | 0.872 | 0.890 | 0.315 | 0.365 | 0.316 | 0.163 | 1 |
| | 25 | Piglet | 1.391 | 1.558 | 1.463 | 0.236 | 0.334 | 0.531 | 0.390 | 0.097 | 2 |
| | 26 | Piglet | 0.466 | 0.228 | 0.459 | 0.274 | 0.374 | 0.541 | 0.412 | 0.160 | 2 |
| | 27 | Fattening pig | 0.519 | 0.240 | 0.484 | 0.354 | 0.737 | 0.765 | 0.774 | 0.760 | 4 |
| | 28 | Fattening pig | 0.634 | 1.007 | 0.770 | 1.021 | 0.369 | 0.438 | 0.438 | 0.227 | 4 |
| | 29 | Sows | 1.775 | 0.845 | 0.761 | 0.693 | 2.011 | 0.409 | 0.306 | 0.455 | 6 |

**TABLE 1** (Continued)

| Samples | No. | Collection | Different proteins, IgG values | | | | Different proteins, IgA values | | | | NTs |
|---|---|---|---|---|---|---|---|---|---|---|---|
| | | | S1 | E | M | N | S1 | E | M | N | |
| | 30 | Piglet | 0.524 | 0.521 | 0.570 | 0.613 | 0.114 | 0.199 | 0.114 | 0.097 | 0 |
| | 31 | Piglet | 0.455 | 0.222 | 0.428 | 0.238 | 0.191 | 0.586 | 0.273 | 0.132 | 0 |
| | 32 | Piglet | 0.914 | 0.716 | 0.676 | 0.749 | 0.555 | 0.890 | 0.388 | 0.213 | 3 |
| | 33 | Piglet | 0.685 | 0.794 | 0.663 | 0.866 | 0.294 | 0.450 | 0.314 | 0.213 | 1 |
| | 34 | Piglet | 0.655 | 0.991 | 0.852 | 1.024 | 0.644 | 0.567 | 0.685 | 0.530 | 3 |
| | 35 | Fattening pig | 0.532 | 0.268 | 0.503 | 0.625 | 0.591 | 0.275 | 0.250 | 0.152 | 3 |
| | 36 | Piglet | 1.218 | 0.483 | 0.805 | 1.003 | 0.179 | 0.206 | 0.165 | 0.138 | 0 |
| | 37 | Piglet | 0.546 | 0.438 | 0.572 | 0.528 | 0.880 | 0.403 | 0.212 | 0.148 | 4 |
| | 38 | Piglet | 0.911 | 1.201 | 0.946 | 0.754 | 0.332 | 0.534 | 0.324 | 0.136 | 2 |
| | 39 | Fattening pig | 2.403 | 1.428 | 1.325 | 1.982 | 1.380 | 0.568 | 0.206 | 0.602 | 4 |
| | 40 | Piglet | 0.894 | 1.103 | 0.805 | 0.514 | 0.532 | 0.407 | 0.254 | 0.130 | 2 |
| | 41 | Piglet | 0.645 | 1.233 | 0.933 | 0.769 | 0.145 | 0.418 | 0.227 | 0.145 | 1 |
| | 42 | Piglet | 1.918 | 1.938 | 1.942 | 2.600 | 0.377 | 0.532 | 0.302 | 0.173 | 2 |
| | 43 | Piglet | 1.003 | 0.799 | 0.768 | 0.856 | 0.363 | 0.759 | 0.505 | 0.294 | 2 |
| | 44 | Piglet | 1.003 | 0.375 | 0.629 | 0.969 | 0.198 | 0.222 | 0.189 | 0.158 | 0 |
| | 45 | Piglet | 0.614 | 0.654 | 0.619 | 0.746 | 0.461 | 0.332 | 0.186 | 0.143 | 3 |
| | 46 | Piglet | 0.995 | 1.224 | 0.924 | 1.132 | 0.325 | 0.596 | 0.348 | 0.177 | 2 |
| Colostrum | 11 | All samples | 0.650 | | | | 0.338 | | | | 2 |
| | 22 | from sows | 0.330 | | | | 0.374 | | | | 2 |
| | 33 | | 2.440 | | | | 2.446 | | | | 8 |
| | 44 | | 0.400 | | | | 0.339 | | | | 2 |
| | 5 | | 0.120 | | | | 0.393 | | | | 2 |
| | 6 | | 0.350 | | | | 0.268 | | | | 1 |
| | 7 | | 0.200 | | | | 0.315 | | | | 1 |
| | 8 | | 0.450 | | | | 0.334 | | | | 2 |
| | 9 | | 0.700 | | | | 0.374 | | | | 2 |
| | 10 | | 1.600 | | | | 2.032 | | | | 6 |
| | 11 | | 1.200 | | | | 1.879 | | | | 4 |
| | 12 | | 1.500 | | | | 2.110 | | | | 5 |
| | 13 | | 0.900 | | | | 1.990 | | | | 3 |
| | 14 | | 0.900 | | | | 2.056 | | | | 6 |
| | 15 | | 1.600 | | | | 1.538 | | | | 5 |
| | 16 | | 0.400 | | | | 0.128 | | | | 1 |
| | 17 | | 0.300 | | | | 0.050 | | | | 1 |
| | 18 | | 0.400 | | | | 0.052 | | | | 2 |
| | 19 | | 0.600 | | | | 0.092 | | | | 2 |
| | 20 | | 0.210 | | | | 0.074 | | | | 1 |
| | 21 | | 0.580 | | | | 0.077 | | | | 2 |
| | 22 | | 0.600 | | | | 0.077 | | | | 1 |
| | 23 | | 0.650 | | | | 0.171 | | | | 1 |
| | 24 | | 0.400 | | | | 0.166 | | | | 1 |
| | 25 | | 0.690 | | | | 0.102 | | | | 1 |
| | 26 | | 0.580 | | | | 0.171 | | | | 1 |
| | 27 | | 0.670 | | | | 0.123 | | | | 1 |
| | 28 | | 0.890 | | | | 0.062 | | | | 1 |
| | 29 | | 0.500 | | | | 1.165 | | | | 3 |
| | 30 | | 1.800 | | | | 1.426 | | | | 4 |
| | 31 | | 1.600 | | | | 1.551 | | | | 4 |
| | 32 | | 1.600 | | | | 1.369 | | | | 3 |

**Plasmid construction.** The PEDV S1, E, M, and N genes of AH2012/12 strain with His tags at the C termini were optimally synthesized and cloned into pcDNA3.1 vector by Nanjing GenScript Biotechnology, and the constructed plasmids were named pPEDV-S1, pPEDV-E, pPEDV-M, and pPEDV-N, respectively.

**Indirect IF.** Indirect IF assays were performed as described before with little modification (34). Briefly, the four plasmids pPEDV-S1, pPEDV-E, pPEDV-M, and pPEDV-N were transfected into 293T cells for 24 h, respectively. The transfected cells were then fixed with 4% formaldehyde for 0.5 h and blocked with phosphate-buffered saline (PBS) containing 5% bovine serum albumin for 2 h. After being blocked, the cells were incubated for 1 h with His-tagged monoclonal antibody (Proteintech Group) diluted 1:1,000. The cells were then washed and incubated with secondary antibody (Cy3-conjugated goat anti-mouse IgG diluted 1:500) for 45 min. Finally, cells were washed and visualized by fluorescence microscopy (Olympus, CKX53).

**Expression and purification of proteins.** HEK 293F cells were seeded at a concentration of $2 \times 10^6$ cells/mL into SMM293-TII medium. After 24 h, the plasmids pPEDV-S1, pPEDV-E, pPEDV-M, and pPEDV-N were transfected with Sinofection transfection reagent (Sino Biological, Inc., China) and SMS 293-SUPI cell culture supplement (Sino Biological, Inc., China). The protein-containing supernatants were collected at 72 h after transfection, and protein was purified using HisTrap HP columns (Cytiva, USA) and an NGC Quest 10 Plus system (Bio-Rad, USA). The purified proteins were desalted and quantified using a Bradford protein concentration detection kit (Beyotime, China).

**SDS-PAGE and immunoblotting.** Purified S1, E, M, and N proteins were separated by SDS-PAGE using 10% separating gel and 5% stacking gel. Gels were stained with Coomassie brilliant blue R250 according to the method of Burnette (34). Immunoblotting was performed as described before (32). Briefly, protein samples separated on gels were transferred onto 0.22-$\mu$m polyvinylidene fluoride (PVDF) membranes (Millipore, Bedford, MA, USA). The membranes were blocked with 5% (wt/vol) nonfat dry milk overnight at 4°C and probed with anti-His (Proteintech Group) primary antibody at 37°C for 1 h. After thorough washing with PBS + 0.1% (vol/vol) Tween 20 (PBST), the membranes were incubated with horseradish peroxidase (HRP)-conjugated secondary antibodies (Proteintech Group) overnight at 4°C. The target protein blots on membranes were developed with an enhanced chemiluminescence detection kit (Thermo Fisher Scientific), and images were acquired using a Tanon 5200 CE Chemi-Image system (Tanon, Shanghai, China).

**Neutralizing antibody titer assays.** Serum or colostrum samples were inactivated at 56°C for 30 min. For fecal samples, 0.1 g was weighed and dissolved in 500 $\mu$L of sterile PBS and centrifuged at 10,000 rpm for 10 min at 4°C. The supernatant was collected and used for determination. Penicillin-streptomycin-amphotericin B solution was added to inactivated serum and colostrum samples and fecal samples. After 2-fold serial dilutions starting at 1:2, sera, feces, or colostrum were mixed with PEDV (200 50% tissue culture infective dose [$TCID_{50}$]/0.1 mL) at an equal volume and coincubated at 37°C for 1 h. The mixture was then inoculated onto the Vero cell monolayers of a 96-well tissue culture plate for 1.5 h at 37°C. After incubation, the mixture was discarded, and the plate was washed three times with DMEM. Next, DMEM containing trypsin (5 $\mu$g/mL), which is needed for virus entry into cells (32), was added to each well, and the plate was incubated for 3 to 5 days at 37°C. Neutralizing antibody titers were expressed as the $log_2$ transformation of the reciprocal highest sera, feces, or colostrum dilution that completely inhibited cytopathic effects (CPEs). Each sample was run in triplicate. The levels of IgG and IgA and NTs of all samples are showed in Table 1.

**ELISA detection.** To detect the IgG and IgA antibodies specific to S1, E, M, or N proteins, indirect ELISA methods were developed in our laboratory. Briefly, microtiter plates were coated with 25, 200, 200, or 200 ng/well of the S1, E, M, or N antigens diluted in coating buffer (50 mM bicarbonate buffer, pH 9.6) and incubated overnight at 4°C. After three washes with PBST, the plates were blocked with 5% powdered skim milk in PBST for 1 h at 37°C and incubated with serum or colostrum samples diluted 1:200 in PBST containing 5% powdered skim milk and fecal samples diluted 1:20 in PBST containing 5% powdered skim milk for 0.5 h at 37°C. After washing, a 1:20,000 diluted peroxidase-conjugated goat anti-porcine IgG (Abcam, UK) or IgA (Abcam, UK) (isotype specific) was added and incubated at 37°C for 0.5 h. The peroxidase reaction was visualized using tetramethylbenzidine-hydrogen peroxide (InnoReagents, HuZhou, China) as the substrate for 20 min at room temperature in the dark and was stopped by adding 2 M sulfuric acid as the stop solution to each well. OD values were measured at 450 nm. Positive, negative, and blank (sterile water) samples were included on each plate, and all clinical and control samples were tested in duplicate.

**Data analysis.** All data analyses and graph plotting were performed using GraphPad Prism (GraphPad Software, La Jolla, CA). The IgG and IgA OD values on reactivity to viral proteins and NTs of all samples were compared by correlation analysis, and the values are presented as correlation and mean $\pm$ standard deviation (SD). $R^2$ values represent correlation coefficients, and $P$ values of $<$0.05 were considered statistically significant.

**Ethics statement.** All animal experiments were performed with the approval of the Jiangsu Academy of Agricultural Sciences Experimental Animal Ethics Committee (NKYVET 2015-0127) and in accordance with relevant guidelines and regulations. All efforts were made to minimize animal suffering and to reduce the number of animals used.

## ACKNOWLEDGMENTS

This work was funded by National Key Research and Development Program of China (grant number 2021YFD1801104), National Natural Science Foundation of China (grant numbers 32272996, 32202823, and 32202787), China Postdoctoral Science Foundation (grant numbers 2022M711398 and 2022M711399), Natural Science Foundation of Jiangsu Province (grant numbers BK20191235, BK20221432, and BK20210158), Jiangsu Agricultural Science and Technology Innovation Fund (grant number CX[21]2038), Innovation Foundation of Jiangsu Academy of Agricultural Sciences (ZX[21]1217), The Special Project of Northern Jiangsu (SZ-LYG202109), and The "JBGS" Project of Seed Industry Revitalization in Jiangsu Province (JBGS[2021]024).

We declare no competing interests.

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
