## [Reviewer comments · Microbiology Spectrum]

Microbiology Spectrum

Correlation between the IgG/IgA antibody response against PEDV structural protein and virus neutralization

Xu Song, Jiali Qian, Chuanhong Wang, Dandan Wang, Junming Zhou, Yongxiang Zhao, Wei Wang, Jizong Li, Rongli Guo, Yunchuan Li, XUEJIAO ZHU, Shanshan Yang, Xuehan Zhang, Baochao Fan, and Bin Li

Corresponding Author(s): Baochao Fan, Jiangsu Academy of Agricultural Sciences

Review Timeline:

Submission Date:	December 23, 2022
Editorial Decision:	February 6, 2023
Revision Received:	March 7, 2023
Accepted:	March 9, 2023

Editor: Biao He

Reviewer(s): The reviewers have opted to remain anonymous.

Transaction Report:

DOI: <https://doi.org/10.1128/spectrum.05233-22>

February 6, 2023

Dr. Baochao Fan
Jiangsu Academy of Agricultural Sciences
Nanjing
China

Re: Spectrum05233-22 (Correlation between the IgG/IgA antibody response against PEDV structural protein and virus neutralization)

Dear Dr. Baochao Fan:

Thank you for submitting this manuscript to Spectrum. The manuscript has been carefully read by three experts with expertise addressed in this study. As you can see below, they raised many concerns. I invite you to thoroughly revise your manuscript based on these comments.

Link Not Available

Sincerely,

Biao He

Journals Department
Reviewer comments:

Reviewer #1 (Comments for the Author):

Xu Song et al have submitted a manuscript entitled "Correlation between the IgG/IgA antibody 1 response against PEDV structural protein and virus neutralization" where they measure antibody titers against PEDV antigens in various samples and determine the neutralization titers of these antibodies against PEDV, and finally analyze correlation between antibody levels and neutralization titers. They find that IgA antibodies against S1 in serum and colostrum highly correlated with neutralization titers. Although the methods and materials used in this work are acceptable and experimental design appears to have no flaws, I do not see anything novel in the data since most of this has been established already. The figures are nicely presented although some the legends require correction. The English in the manuscript needs to be revised as there are many grammatical and

stylistic errors. Further, I do have a few issues with the work as presented.

1. I am not fond of propagating viruses in non-host cells when a host cell line is available, e.g., the authors have used Vero for propagating PEDV. Although the cells are permissive, I believe the replication kinetics are not the same as those in host cells such as PK-15. Line 96 - Please provide a justification of propagating a porcine virus in a monkey cell line. Are the replication kinetics similar? The fact that others have done the same previously is not a justification. Why not use a porcine line, such as PK-15? Similarly, are human cell lines suitable for this work?
2. They have determined neutralization capacity of antibodies against PEDV nucleoprotein antigen. This protein is located beneath the virus envelope. How possible is it that the antibodies penetrate the virion envelope to bind the N protein?
3. Lines 113-119 - What were the authors detecting in this IF assay? I can only guess that they could have been detecting His-tagged PEDV proteins, but how did those viral proteins get there? Please clarify and revise.
4. Line 162 - Does not make sense. You add Ab/virus mixture to cells and then you wash. Was there any incubation period preceding the wash? (Adsorption period?). Please clarify.
5. Fig. 1A please add some detail to describe what the reader is looking at. Besides, in the material and methods section it is said that the detecting antibodies were labeled with FITC, and FITC usually looks green and not red. Please clarify. Fig.1B and C - Are these figures showing the same proteins? Particularly, PEDV-S1 appears larger on C than B. Were these assays run on different gel concentrations or were there modifications in run time? For sure it was not B that was transferred onto C for blotting. Please clarify.
6. Line 195 and Fig 2A - Are the authors asserting that antibodies against nucleocapsid protein penetrated the virions to bind to the NP antigen which is located internally? Please provide a mechanism of how this is happening. My understanding is that the virus used in the NT assays was purified.
7. Line 42 - Revise the English grammar.
8. Line 61 - please include the virus family and genus.
9. Line 74 - revise with "in the"
10. Line 78 - revise "neutralization"
11. Line 79 - revise "infection"
12. Lines - 89-91 - Please revise this sentence.
13. Line 105 - Please be distinctive that colostrum is from sows and not piglets.
14. Line 110- please revise.
15. Line 124 - The proteins were collected from where? Please clarify.
16. Line 133 - remove the parentheses.
17. Lines 158 - 162 - Please revise this sentence for clarity.
18. Line 168 - Please revise the writing of this sentence.
19. Line 219 and Fig 4. - Please correct the legend. It is about S1 and not other viral proteins.
20. Line 227-229 - Do antibodies penetrate virus envelopes?
21. Line 232-233 - What does this mean?

Reviewer #2 (Comments for the Author):

In this study, Song et al. analyze a big set of samples from pigs for reactivity by ELISA against in house produced proteins of the porcine epidemic diarrhea virus (PEDV), a virus with high mortality rates in piglets and therefore of economic relevance. The proteins tested were S1, M, E and N of the most common strain of virus. The main findings of the manuscript are that IgG and IgA OD values to S1 in serum, fecal matter and colostrum samples correlated well with neutralization activity. Interestingly, authors also found similar correlations with the N protein. I think the correlations are clear and the manuscript has potential. However, it needs a lot of work in the writing and the English. I feel the authors can take further advantage of their findings and improve considerably their manuscript.

- The manuscript needs to be carefully revised and many sections rewritten. Regarding this, I have a non-extensive list below of examples, but the text needs further improvements other than my suggestions below:

- I feel authors could significantly improve their discussion. In that sense, what do they think about the correlations with the N protein? Neutralization titers correlating with an internal protein would not be expected, in opposition to a correlation with a spike/envelope protein. What's the authors take on that? Please add a statement or a few sentences in the discussion.
- I found the abstract a bit confusing, so I suggest the authors to improve it by rewriting it. I suggest the author to make the abstract as easy as possible to understand, even for those who have not (or will not) read the manuscript.
- Line 23: "The neutralizing antibody plays pivotal roles in protecting piglets" should read "Neutralizing antibodies play a pivotal role in protecting piglets" or the like.
- Line 25: I would add the word "absorbance" before "values" for enhanced clarity.
- Line 30: "R-squared revealed" should read "R-square values revealed" or the like.
- Line 31: "And the correlations of anti-E or" should read "The correlations of anti-E or"
- Line 51: the term "highly exposed" was not clear to me in this context. I suggest the authors re-write this sentence for enhanced clarity.
- Line 53: "mortality rate of diarrhea piglets" should read "mortality rate in piglets with diarrhea", "mortality rate in piglets" or the

like.

- Line 54: Just a technicality: "was first discovered in the United Kingdom in 1971 and was later found in European countries" should read "was first discovered in the United Kingdom in 1971 and was later found in other European countries", since the UK separated from the European Union but it is still part of Europe.
- Line 61: I suggest removing the "a" from "a positive-sense RNA virus".
- Line 65: "is a principle surface glycoprotein". Did the authors mean "is the main surface glycoprotein"? Please clarify/rewrite for clarity.
- Line 75: "homotypic antibody-producing cells". I am not sure what the authors meant here. Aren't all b-cells homotypic, in the sense that they all make one antibody that binds just one epitope? I suggest removing the word homotypic to avoid confusion.
- Line 79: "after PEDV excitation", perhaps "after PEDV challenge" was meant here. Please correct or clarify.
- Line 83: "et al" should read "et al."
- Line 93: please re-write for clarity.
- Line 96: "was" should read "were".
- Line 100: "preserved in our lab maintained" should read "preserved in our lab were maintained".
- Line 105, 28 and other parts of the manuscript: the plural of serum is serums or sera. Either serum should be changed to "sera" or add "samples" so that it reads "serum samples".
- Line 106: "the pig farms in Jiangsu province" should read "in pig farms in the Jiangsu province".
- Line 110: "which named" should read "which were named".
- Line 114: "Briefly, 293T cells". I would provide more detail here and indicate that the cells were transfected, for completeness.
- Line 124: "the protein was collected". I would indicate here "the supernatant was collected" or "the protein-containing supernatant was collected".
- Line 125: "and purified" I would indicate "and protein was purified".
- Line 130-131: please remove "was performed" for clarity.
- Line 159: "but the Penicillin-Streptomycin-Amphotericin B Solution need to be added" should read "but a Penicillin-Streptomycin-Amphotericin B Solution needed to be added".
- Line 163, 165 and others: While the word "thrice" is correct, it is however quite outdated and in disuse. I advise the authors to change it to "three times".
- Line 168: please define CPEs.
- Line 168: "the IgG, IgA and NTs". I recommend the authors to add more text here: "the IgG, IgA OD values on reactivity to viral proteins and NTs of all samples".
- Line 168-169: ".were showed in Table 1:" should read "were showed in Table 1".
- Line 174: "and $p < 0.05$ " should read "and p values < 0.05 ".
- Line 175: "0.01 considered" should read "0.01 were considered".
- Line 178: please define IFs or better yet, don't abbreviate and use immunofluorescence.
- Line 191, 197, 199 and along the manuscript: authors talk about IgG and IgA values but I think they should refer to them as IgG and IgA OD (or absorbance) values or the like, for enhanced clarity.
- Line 193, 204, etc.: not sure how I feel about the use of the 2^1-2^8 units, perhaps authors could add "(1:2 to 1:256)" after that? Or state clearly that those are the log2 transformation of the reciprocal dilutions?
- Line 198: I believe "IgG OD values" should read "IgA OD values".
- Line 222: "epitope" should read "epitopes".
- Line 227: "whatever" should read "either".
- Line 227: please add "matter" or the like after "fecal".
- Lines 229-230: this should read "Several non-neutralizing epitopes have been identified".
- Line 239: "inhibition PEDV" should read "inhibition of PEDV".
- Line 244: I suggest using "weak" instead of "low" here (weak correlations instead of "the low" correlations).
- Line 256: "monitor infected pigs". Wouldn't this be "pigs with prior infection"? or would neutralization really develop quickly during the infection?
- Line 279: "rapidly detect the neutralization capacity". Was it meant here that it can be used as a tool to predict the neutralization capacity of a sample? Please clarify.
- Line 183: please leave a space between number and units.
- Line 390: "was" should read "were".
- Line 391: I suggest stating "eukaryotic expression plasmid" here.
- Line 392: I would like to suggest the authors to use "analysis of proteins purified according to materials and methods" instead of "analysis of purification of proteins".
- Line 211: "obtained by serum" should read "obtained with serum". In same line, "reveals" should read "reveal".
- Line 213-216: found this section a bit confusing. I think authors could re-write for enhanced clarity. Perhaps something along the lines of "because the highest correlation... and NTs was focused on S1, we decided to test now reactivity to S1 in colostrum samples."

- Figure 1B and C: I have some concerns regarding the purity of the purified proteins, more details on the validation is required. I am having some trouble interpreting some apparent differences in molecular weight between the two images. How would the authors explain this?

- Lines 229-234: this section was confusing. Not sure if I understood it well but I am afraid I could not see the relationship

between correlations of IgA OD values and NTs with the fact that some epitopes might not have been identified in the N-protein. Also, how would this be related with the antibodies co-existing for longer time? If the N-protein is not on the surface of the virus, I don't see how antibodies to it could neutralize infectivity, but a correlation may exist between neutralization and N protein reactivity since that reactivity is generated early in infection and perhaps the better response, the better neutralization is generated later on.

- Line 179-180: here the authors state that the fluorescence was largely displayed at 24h post-transfection but they did not show other time points to compare, so I suggest they re-write this sentence to better describe their findings ("high levels of expression were seen at 24h post-transfection" or the like).

- Line 104: I think a bit more detail on how the samples were collected in this section of Materials and Methods would be helpful. For instance, there is not much indication on how or when colostrum was obtained. Only later on, in the discussion, it is indicated that they were collected on day 1-3 post-delivery.

- How many samples were from piglets and how many were from sows/pigs? Please indicate.

- Were the samples matched? As in were for instance the serum samples taken from the same pigs as the fecal ones? Please indicate so.

- Line 150: Were the anti-IgG and anti-IgA used isotype specific, i.e. would the anti-IgG not bind IgA and the other way around? Please indicate since I think this is relevant to the study.

- Line 41: "enterovirus PEDV". Since PEDV is a coronavirus, I suggest the authors remove the word "enterovirus" which would induce confusion. In line with this I feel the authors could add a bit more information on PEDV in the introduction, like the fact that adult pigs have more chances of survival than piglets when infected, how the virus is transmitted, info on what family of virus the PEDV belongs could also be added around line 64.

- Line 41-42: authors could think of adding references after this statement for completeness.

- I am positively surprised that running the neutralization assays directly with fecal samples did not contaminate the cultures, so I am assuming that the Pen/Strep/Ampho worked well and the fact that washes were performed afterwards helped remove bacteria. Out of curiosity, why was trypsin added to the washing buffer and maintenance medium of the neutralization assays?

- Why was colostrum only tested with S protein?

- Figure 1: Adding a bar as a marker/reference for size in Fig 1A would help.

- I think the line charts need further explanation. What is the x-axis? It reads "correlation between IgG-ELISA and neutralization in serum" but it looks to me more like the number of samples. This should be clarified and what is plotted on what axis (left or right) should also be mentioned in the legends.

Reviewer #4 (Comments for the Author):

In this manuscript, correlation between the IgG/IgA antibody response against PEDV structural protein and virus neutralization was reported by Song et al in serum, fecal and colostrum samples, confirming the highest correlation between the neutralizing antibody titers (NTs) and the IgA to PEDV S1 protein. This study proposed that the diagnostic method for anti-S1 IgA can be used for the determination of viral infection and evaluation of vaccine immunization. In general, abundant data was generated and presented in this manuscript, which might have important guiding implications in the surveillance of viral infections, as well as in the evaluation of vaccine immune protection. However, some concerns need to be addressed as below:

1. There are many grammatical errors in this manuscript, such as lines 34-36, lines 64-67, and lines 130-131. The English should be well polished.
2. Abbreviations should be defined at first mention in the main body part, and then subsequently used throughout the manuscript, such as "PED" in line 51 and "CPEs" in line 168.
3. The authors described that "PEDV can be genetically separated into two genotypes: GI (classic) and GII (field epidemic). Each genotype can be further divided into the subgenotypes GIa and GIb and GIIa and GIIb, respectively" in lines 57-59. How about S-INDEL?
4. In this study, results showed that both the IgG and the IgA levels of S1 has the highest correlation with NTs compared to other proteins, but the authors did not explore the correlation between the NTs and the IgG/IgA to PEDV S2 protein. How about the results to S2 protein? Please add the description in discussion.
5. In line 216, "thirty-to" should be "thirty-two", please correct.

Staff Comments:

Preparing Revision Guidelines

Please return the manuscript within 60 days; if you cannot complete the modification within this time period, please contact me. If you do not wish to modify the manuscript and prefer to submit it to another journal, please notify me of your decision immediately so that the manuscript may be formally withdrawn from consideration by Microbiology Spectrum.

In this manuscript, correlation between the IgG/IgA antibody response against PEDV structural protein and virus neutralization was reported by Song et al in serum, fecal and colostrum samples, confirming the highest correlation between the neutralizing antibody titers (NTs) and the IgA to PEDV S1 protein. This study proposed that the diagnostic method for anti-S1 IgA can be used for the determination of viral infection and evaluation of vaccine immunization. In general, abundant data was generated and presented in this manuscript, which might have important guiding implications in the surveillance of viral infections, as well as in the evaluation of vaccine immune protection. However, some concerns need to be addressed as below:

1. There are many grammatical errors in this manuscript, such as lines 34-36, lines 64-67, and lines 130-131. The English should be well polished.
2. Abbreviations should be defined at first mention in the main body part, and then subsequently used throughout the manuscript, such as “PED” in line 51 and “CPEs” in line 168.
3. The authors described that “PEDV can be genetically separated into two genotypes: GI (classic) and GII (field epidemic). Each genotype can be further divided into the subgenotypes GIa and GIb and GIIa and GIIb, respectively” in lines 57-59. How about S-INDEL?
4. In this study, results showed that both the IgG and the IgA levels of S1 has the highest correlation with NTs compared to other proteins, but the authors did not explore the correlation between the NTs and the IgG/IgA to PEDV S2 protein. How about the results to S2 protein? Please add the description in discussion.
5. In line 216, “thirty-to” should be “thirty-two”, please correct.

Dear Editors,

Based on the comments of reviewers, we made extensive modification on the original manuscript. The responses point to point to reviewers' comments are as follows (blue words):

Reviewer comments:

Reviewer #1 (Comments for the Author):

Xu Song et al have submitted a manuscript entitled "Correlation between the IgG/IgA antibody 1 response against PEDV structural protein and virus neutralization" where they measure antibody titers against PEDV antigens in various samples and determine the neutralization titers of these antibodies against PEDV, and finally analyze correlation between antibody levels and neutralization titers. They find that IgA antibodies against S1 in serum and colostrum highly correlated with neutralization titers. Although the methods and materials used in this work are acceptable and experimental design appears to have no flaws, I do not see anything novel in the data since most of this has been established already. The figures are nicely presented although some the legends require correction. The English in the manuscript needs to be revised as there are many grammatical and stylistic errors. Further, I do have a few issues with the work as presented.

Reply: Thank you very much for your comments. This study was the first to compare different PEDV structural proteins antigens, and explicitly confirmed that the highest correlation between NTs and the IgA to PEDV S1 protein. The entire manuscript has been extensively revised in language and grammar, and new analysis and discussion has been added, resulting in an overall improvement. I hope that the revised manuscript will meet with your approval. Thanks again.

1. I am not fond of propagating viruses in non-host cells when a host cell line is available, e.g., the authors have used Vero for propagating PEDV. Although the cells are permissive, I believe the replication kinetics are not the same as those in host cells such as PK-15. Line 96 - Please provide a justification of propagating a porcine virus in a monkey cell line. Are the replication kinetics similar? The fact that others have done the same previously is not a justification. Why not use a porcine line, such as PK-15? Similarly, are human cell lines suitable for this work?

Reply: Thank you very much for your comments. For PEDV, Vero cells are very commonly used as a cell line for viral proliferation and the titer of PEDV proliferation in Vero cells will be higher compared to current porcine cell lines such as PK-15 and IPEC-J2. However, when performing studies on innate immune regulation, the porcine cell lines are the most formal and commonly used. And the replication kinetics of PEDV in different cell lines are usually different. Vero cells are also the most commonly used cell line when performing neutralizing antibody assays (doi: 10.3389/fimmu.2021.785293; doi: 10.3390/ani13040757). Therefore, in this study, Vero cells were used for the proliferation and neutralizing antibody assays. Of course, a suitable porcine-derived cell line will make the data more accurate and convincing, and this is an area that needs to be overcome in future studies.

2. They have determined neutralization capacity of antibodies against PEDV

nucleoprotein antigen. This protein is located beneath the virus envelope. How possible is it that the antibodies penetrate the virion envelope to bind the N protein?

Reply: Thanks for this comment. We analyzed the correlation between IgA antibody levels against N and S1 proteins, and found that serum and fecal samples with high N-IgA levels had equally high S-IgA levels, thus showing high neutralizing activity. Further analysis revealed a higher correlation between IgA antibody levels for the N and S proteins compared to M and E. Such results suggest that it is likely still the IgA against the S protein that exert the neutralizing activity of the virus. The new analyses and descriptions have been added to the Results and Discussion sections. (Lines 150-161, 171-183, and Fig 5)

Fig. 5. The correlations of NTs with IgG (A) or IgA (B) OD values for S1 in colostrum samples. The correlations of the IgA OD values of E, M, N to the IgA OD values of S1 in serum (A) and fecal (B) samples. The correlations were shown in scatter plots. R² values represent the correlation coefficients.

3. Lines 113-119 - What were the authors detecting in this IF assay? I can only guess that they could have been detecting His-tagged PEDV proteins, but how did those viral proteins get there? Please clarify and revise.

Reply: We are sorry for the incomplete description. The IFs was used to detect the expressions of His-tagged PEDV S1, E, M and N proteins. The relevant description has been added in the revised manuscript (Lines 260-268).

4. Line 162 - Does not make sense. You add Ab/virus mixture to cells and then you wash. Was there any incubation period preceding the wash? (Adsorption period?). Please clarify.

Reply: Thanks for this comment. The mixture will be incubated at 37°C for 1.5 h preceding the wash. This has been added in the revised manuscript (Lines 298-299).

5. Fig. 1A please add some detail to describe what the reader is looking at. Besides, in the material and methods section it is said that the detecting antibodies were labeled with FITC, and FITC usually looks green and not red. Please clarify. Fig.1B and C - Are these figures showing the same proteins? Particularly, PEDV-S1 appears larger on

C than B. Were these assays run on different gel concentrations or were there modifications in run time? For sure it was not B that was transferred onto C for blotting. Please clarify.

Reply: Thanks for this comment. The details and modifications of Fig. 1A have been added in the legend and Material and Methods (Lines 260-268, 450-452). For Figure 1B and 1C, the S1 protein is approximately 100 kDa in size and a reversibility check revealed that another purified partial S protein was incorrectly used in Fig. 1C. This protein is of lower purity, so the assays in this manuscript were performed using the higher purity S1 protein. The correct figure has been replaced in the revised manuscript. We are very sorry for the mistake. We hope that this response will meet with your approval. (Lines 112-113) and Fig 1.

6. Line 195 and Fig 2A - Are the authors asserting that antibodies against nucleocapsid protein penetrated the virions to bind to the NP antigen which is located internally? Please provide a mechanism of how this is happening. My understanding is that the virus used in the NT assays was purified.

Reply: Thanks for this comment. The neutralizing activity of the antibody against N protein has been analyzed and discussed in the revised manuscript (Lines 150-161, 171-183, and Fig 5).

7. Line 42 - Revise the English grammar.

Reply: Thanks for this comment. This has been modified in the revised manuscript (Line 46).

8. Line 61 - please include the virus family and genus.

Reply: The relevant description has been added in the revised manuscript (Line 70).

9. Line 74 - revise with "in the"

Reply: This has been modified in the revised manuscript (Line 84).

10. Line 78 - revise "neutralization"

Reply: This has been modified in the revised manuscript (Line 88).

11. Line 79 - revise "infection"

Reply: This has been modified in the revised manuscript (Line 89).

12. Lines - 89-91 - Please revise this sentence.

Reply: Thanks for this comment. The sentence has been modified in the revised manuscript (Lines 99-101).

13. Line 105 - Please be distinctive that colostrum is from sows and not piglets.
Reply: Thanks for this comment. This has been modified in the revised manuscript (Lines 116-122).
14. Line 110- please revise.
Reply: This has been modified in the revised manuscript (Line 126).
15. Line 124 - The proteins were collected from where? Please clarify.
Reply: The relevant description has been added in the revised manuscript (Lines 247-253).
16. Line 133 - remove the parentheses.
Reply: This has been modified in the revised manuscript (Line 283).
17. Lines 158 - 162 - Please revise this sentence for clarity.
Reply: Thanks for this comment. The sentence has been revised in the revised manuscript (Lines 292-298).
18. Line 168 - Please revise the writing of this sentence.
Reply: Thanks for this comment. The sentence has been revised in the revised manuscript (Line 305).
19. Line 219 and Fig 4. - Please correct the legend. It is about S1 and not other viral proteins.
Reply: Thanks for this comment. The legend has been revised in the revised manuscript (Line 470).
20. Line 227-229 - Do antibodies penetrate virus envelopes?
Reply: Thanks for this comment. The neutralizing activity of the antibody against N protein has been analyzed and discussed in the revised manuscript (Lines 150-161, 171-183, and Fig 5).
21. Line 232-233 - What does this mean?
Reply: Thanks for this comment. The sentence was replaced with an alternative description (Lines 170-173).

Reviewer #2 (Comments for the Author)

In this study, Song et al. analyze a big set of samples from pigs for reactivity by ELISA against in house produced proteins of the porcine epidemic diarrhea virus (PEDV), a virus with high mortality rates in piglets and therefore of economic relevance. The proteins tested were S1, M, E and N of the most common strain of virus. The main findings of the manuscript are that IgG and IgA OD values to S1 in serum, fecal matter and colostrum samples correlated well with neutralization activity. Interestingly, authors also found similar correlations with the N protein. I think the correlations are clear and the manuscript has potential. However, it needs a lot of work in the writing and the English. I feel the authors can take further advantage of their findings and improve considerably their manuscript.

Reply: Thank you for your affirmation and comments. The writing and grammar of the whole manuscript have been revised. And new analysis and discussion has been added, resulting in an overall improvement. I hope that the revised manuscript will meet with your approval. Thanks again.

1.The manuscript needs to be carefully revised and many sections rewritten.

Regarding this, I have a non-extensive list below of examples, but the text needs further improvements other than my suggestions below:

Reply: Thank you very much for your exhaustive revisions and sorry also for the mistakes in our work. The writing and grammar of the whole manuscript have been revised with very careful modifications. And the quality of the whole manuscript has been greatly improved. Thanks again.

• I feel authors could significantly improve their discussion. In that sense, what do they think about the correlations with the N protein? Neutralization titers correlating with an internal protein would not be expected, in opposition to a correlation with a spike/envelope protein. What's the authors take on that? Please add a statement or a few sentences in the discussion.

Reply: Thanks for this comment. We analyzed the correlation between IgA antibody levels against N and S1 proteins, and found that serum and fecal samples with high N-IgA levels had equally high S-IgA levels, thus showing high neutralizing activity. Further analysis revealed a higher correlation between IgA antibody levels for the N and S proteins compared to M and E. Such results suggest that it is likely still the IgA against the S protein that exert the neutralizing activity of the virus. The new analyses and descriptions have been added to the Results and Discussion sections. (Lines 150-161, 171-183, and Fig 5)

Fig. 5. The correlations of NTs with IgG (A) or IgA (B) OD values for S1 in colostrum samples. The correlations of the IgA OD values of E, M, N to the IgA OD values of S1 in serum (A) and fecal (B) samples. The correlations were shown in scatter plots. R2 values represent the correlation coefficients.

• I found the abstract a bit confusing, so I suggest the authors to improve it by rewriting it. I suggest the author to make the abstract as easy as possible to understand, even for those who have not (or will not) read the manuscript.

Reply: Thanks for this comment. The abstract has been modified (Lines 22-40).

• Line 23: "The neutralizing antibody plays pivotal roles in protecting piglets" should read "Neutralizing antibodies play a pivotal role in protecting piglets" or the like.

Reply: This has been modified in the revised manuscript (Lines 24-25).

- Line 25: I would add the word "absorbance" before "values" for enhanced clarity.

Reply: This has been added in the revised manuscript (Line 27).

- Line 30: "R-squared revealed" should read "R-square values revealed" or the like.

Reply: The related statements have been modified in the revised manuscript (Line 32).

- Line 31: "And the correlations of anti-E or" should read "The correlations of anti-E or"

Reply: This has been modified in the revised manuscript (Line 34).

- Line 51: the term "highly exposed" was not clear to me in this context. I suggest the authors re-write this sentence for enhanced clarity.

Reply: We are sorry for the description. This sentence has been modified for clarity (Line 53).

- Line 53: "mortality rate of diarrhea piglets" should read "mortality rate in piglets with diarrhea", "mortality rate in piglets" or the like.

Reply: Thanks for this comment. This has been modified in the revised manuscript (Line 55-56).

- Line 54: Just a technicality: "was first discovered in the United Kingdom in 1971 and was later found in European countries" should read "was first discovered in the United Kingdom in 1971 and was later found in other European countries", since the UK separated from the European Union but it is still part of Europe.

Reply: Thanks, this has been modified in the revised manuscript (Line 60-61).

- Line 61: I suggest removing the "a" from "a positive-sense RNA virus".

Reply: The relevant description has been removed in the revised manuscript (Line 71).

- Line 65: "is a principle surface glycoprotein". Did the authors mean "is the main surface glycoprotein"? Please clarify/rewrite for clarity.

Reply: Thanks for this comment. This has been modified in the revised manuscript (Lines 75-76).

- Line 75: "homotypic antibody-producing cells". I am not sure what the authors meant here. Aren't all b-cells homotypic, in the sense that they all make one antibody that binds just one epitope? I suggest removing the word homotypic to avoid confusion.

Reply: Thanks for this comment. The word has been removed for clarity (Line 85).

- Line 79: "after PEDV excitation", perhaps "after PEDV challenge" was meant here. Please correct or clarify.

Reply: We are sorry for the unclear description. This has been modified in the revised manuscript (Line 89).

- Line 83: "et al" should read "et al."

Reply: This has been modified in the revised manuscript (Line 91).

- Line 93: please re-write for clarity.

Reply: This has been modified in the revised manuscript (Lines 103-104).

- Line 96: "was" should read "were".

Reply: This has been modified in the revised manuscript (Line 238).

- Line 100: "preserved in our lab maintained" should read "preserved in our lab were maintained".

Reply: This has been modified in the revised manuscript (Lines 242-243).

• Line 105, 28 and other parts of the manuscript: the plural of serum is serums or sera. Either serum should be changed to "sera" or add "samples" so that it reads "serum samples".

Reply: This has been modified in the revised manuscript (Line 247 and other parts of the manuscript).

• Line 106: "the pig farms in Jiangsu province" should read "in pig farms in the Jiangsu province".

Reply: Thanks for this comment. This has been modified in the revised manuscript (Lines 248-249).

• Line 110: "which named" should read "which were named".

Reply: Thanks for this comment. This has been modified in the revised manuscript (Line 257).

• Line 114: "Briefly, 293T cells". I would provide more detail here and indicate that the cells were transfected, for completeness.

Reply: Thanks for this comment. The more detail was added in the revised manuscript (Lines 260-268).

• Line 124: "the protein was collected". I would indicate here "the supernatant was collected" or "the protein-containing supernatant was collected".

Reply: Thanks for this comment. This has been modified in the revised manuscript (Lines 273-274).

• Line 125: "and purified" I would indicate "and protein was purified".

Reply: Thanks for this comment. This has been modified in the revised manuscript (Line 274).

• Line 130-131: please remove "was performed" for clarity.

Reply: Thanks for this comment. The words have been removed (Line 281).

• Line 159: "but the Penicillin-Streptomycin-Amphotericin B Solution need to be added" should read "but a Penicillin-Streptomycin-Amphotericin B Solution needed to be added".

Reply: Thanks for this comment. This has been modified in the revised manuscript (Lines 294-296).

• Line 163, 165 and others: While the word "thrice" is correct, it is however quite outdated and in disuse. I advise the authors to change it to "three times".

Reply: Thanks for this comment. This has been modified in the revised manuscript (Line 300).

• Line 168: please define CPEs.

Reply: This has been added in the revised manuscript (Line 304).

• Line 168: "the IgG, IgA and NTs". I recommend the authors to add more text here: "the IgG, IgA OD values on reactivity to viral proteins and NTs of all samples".

Reply: The relevant description has been added in the revised manuscript (Lines 325-326).

• Line 168-169: ".were showed in Table 1:" should read "were showed in Table 1".

Reply: This has been modified in the revised manuscript (Line 305).

• Line 174: "and p<0.05" should read "and p values <0.05".

Reply: This has been modified in the revised manuscript (Line 328).

- Line 175: "0.01 considered" should read "0.01 were considered".

Reply: This has been deleted in the revised manuscript for a more accurate description.

- Line 178: please define IFs or better yet, don't abbreviate and use immunofluorescence.

Reply: This has been added in the revised manuscript (Lines 107-108).

- Line 191, 197, 199 and along the manuscript: authors talk about IgG and IgA values but I think they should refer to them as IgG and IgA OD (or absorbance) values or the like, for enhanced clarity.

Reply: We are sorry for the lack of clarity. This has been modified in the revised manuscript (Lines 121, 128, 131, and other parts of the manuscript).

- Line 193, 204, etc.: not sure how I feel about the use of the 2^1 - 2^8 units, perhaps authors could add "(1:2 to 1:256)" after that? Or state clearly that those are the log₂ transformation of the reciprocal dilutions?

Reply: This has been modified in the revised manuscript (Line 303).

- Line 198: I believe "IgG OD values" should read "IgA OD values".

Reply: We are sorry for the mistake. This has been modified in the revised manuscript (Line 128).

- Line 222: "epitope" should read "epitopes".

Reply: This has been modified in the revised manuscript (Line 163).

- Line 227: "whatever" should read "either".

Reply: This has been modified in the revised manuscript (Line 168).

- Line 227: please add "matter" or the like after "fecal".

Reply: Thanks for this comment. This has been added in the revised manuscript (Line 168).

- Lines 229-230: this should read "Several non-neutralizing epitopes have been identified".

Reply: Thanks for this comment. This has been modified in the revised manuscript (Line 178).

- Line 239: "inhibition PEDV" should read "inhibition of PEDV".

Reply: Thanks for this comment. This has been modified in the revised manuscript (Line 188).

- Line 244: I suggest using "weak" instead of "low" here (weak correlations instead of "the low" correlations).

Reply: Thanks for this comment. This has been modified in the revised manuscript (Line 193).

- Line 256: "monitor infected pigs". Wouldn't this be "pigs with prior infection"? or would neutralization really develop quickly during the infection?

Reply: Thanks for this comment. This has been modified in the revised manuscript (Line 205).

- Line 279: "rapidly detect the neutralization capacity". Was it meant here that it can be used as a tool to predict the neutralization capacity of a sample? Please clarify.

Reply: Thanks for this comment. This has been modified in the revised manuscript

(Lines 233-234).

- Line 183: please leave a space between number and units.

Reply: This has been modified in the revised manuscript (Line 113).

- Line 390: "was" should read "were".

Reply: This has been modified in the revised manuscript (Line 451).

- Line 391: I suggest stating "eukaryotic expression plasmid" here.

Reply: Thanks for this comment. This has been added in the revised manuscript (Lines 451-452).

- Line 392: I would like to suggest the authors to use "analysis of proteins purified according to materials and methods" instead of "analysis of purification of proteins".

Reply: Thanks for this comment. This has been modified in the revised manuscript (Line 452).

- Line 211: "obtained by serum" should read "obtained with serum". In same line, "reveals" should read "reveal".

Reply: This has been modified in the revised manuscript (Line 141).

- Line 213-216: found this section a bit confusing. I think authors could re-write for enhanced clarity. Perhaps something along the lines of "because the highest correlation... and NTs was focused on S1, we decided to test now reactivity to S1 in colostrum samples.

Reply: Thanks for this comment. This has been re-written in the revised manuscript (Lines 144-145).

- Figure 1B and C: I have some concerns regarding the purity of the purified proteins, more details on the validation is required. I am having some trouble interpreting some apparent differences in molecular weight between the two images. How would the authors explain this?

Reply: Thanks for this comment. For Figure 1B and 1C, the S1 protein is approximately 100 kDa in size and a reversibility check revealed that another purified partial S protein was incorrectly used in Fig. 1C. This protein is of lower purity, so the assays in this manuscript were performed using the higher purity S1 protein. The correct figure has been replaced in the revised manuscript. We are very sorry for the mistake. We hope that this response will meet with your approval. (Line 113) and Fig 1.

- Lines 229-234: this section was confusing. Not sure if I understood it well but I am afraid I could not see the relationship between correlations of IgA OD values and NTs with the fact that some epitopes might not have been identified in the N-protein. Also, how would this be related with the antibodies co-existing for longer time? If the N-protein is not on the surface of the virus, I don't see how antibodies to it could neutralize infectivity, but a correlation may exist between neutralization and N protein reactivity since that reactivity is generated early in infection and perhaps the better response, the better neutralization is generated later on.

Reply: Thanks for this comment. The neutralizing activity of the antibody against N protein has been analyzed and discussed in the revised manuscript (Lines 150-161, 171-183, and Fig 5).

- Line 179-180: here the authors state that the fluorescence was largely displayed at 24h post-transfection but they did not show other time points to compare, so I suggest they re-write this sentence to better describe their findings ("high levels of expression were seen at 24h post-transfection" or the like).

Reply: Thanks for this comment. This has been re-written in the revised manuscript (Lines 107-109).

- Line 104: I think a bit more detail on how the samples were collected in this section of Materials and Methods would be helpful. For instance, there is not much indication on how or when colostrum was obtained. Only later on, in the discussion, it is indicated that they were collected on day 1-3 post-delivery.

Reply: Thanks for this comment. The relevant description has been added in the revised manuscript (Lines 247-253 and Table 1).

- How many samples were from piglets and how many were from sows/pigs? Please indicate.

Reply: Thanks for this comment. The relevant description has been added in the revised manuscript (Lines 247-253 and Table 1).

- Were the samples matched? As in were for instance the serum samples taken from the same pigs as the fecal ones? Please indicate so.

Reply: Thanks for this comment. The samples were collected at different times and places, and they did not match.

- Line 150: Were the anti-IgG and anti-IgA used isotype specific, i.e. would the anti-IgG not bind IgA and the other way around? Please indicate since I think this is relevant to the study.

Reply: We are sorry for the unclear description. The goat anti-porcine IgG and IgA are isotype specific. This has been added in the revised manuscript (Line 317).

- Line 41: "enterovirus PEDV". Since PEDV is a coronavirus, I suggest the authors remove the word "enterovirus" which would induce confusion. In line with this I feel the authors could add a bit more information on PEDV in the introduction, like the fact that adult pigs have more chances of survival than piglets when infected, how the virus is transmitted, info on what family of virus the PEDV belongs could also be added around line 64.

Reply: Thanks for this comment. This comment is very reasonable. The word "enterovirus" has been removed and relevant information is added in the revised

manuscript (Lines 45, 57-60).

- Line 41-42: authors could think of adding references after this statement for completeness.

Reply: Thanks for this comment. The references have been added in the revised manuscript (Line 46).

- I am positively surprised that running the neutralization assays directly with fecal samples did not contaminate the cultures, so I am assuming that the Pen/Strep/Ampho worked well and the fact that washes were performed afterwards helped remove bacteria. Out of curiosity, why was trypsin added to the washing buffer and maintenance medium of the neutralization assays?

Reply: Thanks for this comment. The accurate description of the method of neutralization assays have been added in the revised manuscript (Lines 292-301).

- Why was colostrum only tested with S protein?

Reply: Thanks for this comment. Based on previous research results, there is the highest correlation between antibody levels to S1 and NTs. So, we decided to test now reactivity to S1 in colostrum samples. The description has been added in the revised manuscript (Lines 144-145).

- Figure 1: Adding a bar as a marker/reference for size in Fig 1A would help.

Reply: Thanks for this comment. The bar has been changed (Fig. 1A; Line 461).

- I think the line charts need further explanation. What is the x-axis? It reads "correlation between IgG-ELISA and neutralization in serum" but it looks to me more like the number of samples. This should be clarified and what is plotted on what axis (left or right) should also be mentioned in the legends.

Reply: Thanks for this comment. The annotations of x-axis in line charts have been changed, and the descriptions have been added in the figure legends. (Fig. 2, 3 4; Lines 448-461, 464-467, 471-474).

Reviewer #4 (Comments for the Author):

In this manuscript, correlation between the IgG/IgA antibody response against PEDV structural protein and virus neutralization was reported by Song et al in serum, fecal and colostrum samples, confirming the highest correlation between the neutralizing antibody titers (NTs) and the IgA to PEDV S1 protein. This study proposed that the diagnostic method for anti-S1 IgA can be used for the determination of viral infection and evaluation of vaccine immunization. In general, abundant data was generated and presented in this manuscript, which might have important guiding implications in the surveillance of viral infections, as well as in the evaluation of vaccine immune protection. However, some concerns need to be addressed as below:

Reply: Thank you for your affirmation and comments. The writing and grammar of the whole manuscript have been revised. And new analysis and discussion has been added, resulting in an overall improvement. I hope that the revised manuscript will meet with your approval. Thanks again.

1. There are many grammatical errors in this manuscript, such as lines 34-36, lines 64-67, and lines 130-131. The English should be well polished.

Reply: Thanks for this comment. The writing and grammar of the whole manuscript

have been revised with very careful modifications. And the quality of the whole manuscript has been greatly improved (Lines 36-40, 74-77, 280-281).

2. Abbreviations should be defined at first mention in the main body part, and then subsequently used throughout the manuscript, such as "PED" in line 51 and "CPEs" in line 168.

Reply: Thanks for this comment. PED and CPEs has been defined in the revised manuscript (Lines 55, 304). The whole manuscript has been checked and modified.

3. The authors described that "PEDV can be genetically separated into two genotypes: GI (classic) and GII (field epidemic). Each genotype can be further divided into the subgenotypes GIa and GIb and GIIa and GIIb, respectively" in lines 57-59. How about S-INDEL?

Reply: Thanks for this comment. The S-INDEL related contents has been added in the revised manuscript (Lines 63-67).

4. In this study, results showed that both the IgG and the IgA levels of S1 has the highest correlation with NTs compared to other proteins, but the authors did not explore the correlation between the NTs and the IgG/IgA to PEDV S2 protein. How about the results to S2 protein? Please add the description in discussion.

Reply: Thanks for this comment. The description has been added in the discussion (Lines 217-223).

5. In line 216, "thirty-to" should be "thirty-two", please correct.

Reply: We are sorry for the wrong description. The word has been modified in the revised manuscript (Line 145).

The revised manuscript with modifications with yellow highlighting is enclosed. We hope that it can be affirmed. I am looking forward to hearing from you.

Sincerely.

March 9, 2023

Dr. Baochao Fan
Jiangsu Academy of Agricultural Sciences
Nanjing
China

Re: Spectrum05233-22R1 (Correlation between the IgG/IgA antibody response against PEDV structural protein and virus neutralization)

Dear Dr. Baochao Fan:

I am glad to let you know that your manuscript has been accepted, and I am forwarding it to the ASM Journals Department for publication. You will be notified when your proofs are ready to be viewed.

Sincerely,

Biao He
Editor, Microbiology Spectrum
